# The tumour suppressor CYLD regulates the p53 DNA damage response

Vanesa Fernández-Majada[1,†], Patrick-Simon Welz[1,†], Maria A. Ermolaeva[1,2,†], Michael Schell[1,†], Alexander Adam[3], Felix Dietlein[4], David Komander[5], Reinhard Büttner[3], Roman K. Thomas[6], Björn Schumacher[1,2] & Manolis Pasparakis[1]

The tumour suppressor CYLD is a deubiquitinase previously shown to inhibit NF-κB, MAP kinase and Wnt signalling. However, the tumour suppressing mechanisms of CYLD remain poorly understood. Here we show that loss of CYLD catalytic activity causes impaired DNA damage-induced p53 stabilization and activation in epithelial cells and sensitizes mice to chemical carcinogen-induced intestinal and skin tumorigenesis. Mechanistically, CYLD interacts with and deubiquitinates p53 facilitating its stabilization in response to genotoxic stress. Ubiquitin chain-restriction analysis provides evidence that CYLD removes K48 ubiquitin chains from p53 indirectly by cleaving K63 linkages, suggesting that p53 is decorated with complex K48/K63 chains. Moreover, CYLD deficiency also diminishes CEP-1/p53-dependent DNA damage-induced germ cell apoptosis in the nematode *Caenorhabditis elegans*. Collectively, our results identify CYLD as a deubiquitinase facilitating DNA damage-induced p53 activation and suggest that regulation of p53 responses to genotoxic stress contributes to the tumour suppressor function of CYLD.

[1] Institute for Genetics, Cologne Excellence Cluster on Cellular Stress Responses in Aging-Associated Diseases (CECAD) and Centre for Molecular Medicine (CMMC), University of Cologne, Joseph-Stelzmann-Straβe 26, Cologne 50931, Germany. [2] Institute for Genome Stability in Ageing and Disease, Cologne Excellence Cluster for Cellular Stress Responses in Ageing-Associated Diseases (CECAD) and Centre for Molecular Medicine (CMMC), University of Cologne, Joseph-Stelzmann-Straβe 26, Cologne 50931, Germany. [3] Institute of Pathology, University Hospital Cologne, Kerpener Straβe 62, Cologne 50937, Germany. [4] Department I of Internal Medicine, University Hospital of Cologne, Weyertal 115B, Cologne 50931, Germany. [5] Medical Research Council Laboratory of Molecular Biology, Francis Crick Avenue, Cambridge CB2 0QH, UK. [6] Department of Translational Genomics, Center of Integrated Oncology Cologne-Bonn, Medical Faculty, University of Cologne, Weyertal 115b, Cologne 50931, Germany. † Present addresses: Institute for Bioengineering of Catalonia (IBEC), Barcelona 08028, Spain (V.F.-M.); Institute for Research in Biomedicine (IRB), Barcelona 08028, Spain (P.-S.W.); Leibniz Institute for Age Research-Fritz Lipmann Institute, Jena D-07745, Germany (M.A.E.); Cenibra GmbH, Bramsche 49565, Germany (M.S.). Correspondence and requests for materials should be addressed to M.P. (email: pasparakis@uni-koeln.de).

CYLD is a tumour suppressor originally identified as the gene mutated in familial cylindromatosis, a disease characterized by the development of benign skin tumours arising from ectodermal appendages[1]. Subsequently, decreased CYLD expression and mutations of the *CYLD* genomic locus have been reported in a number of different cancers including colon, hepatocellular and myeloid malignancies, suggesting that CYLD exhibits broad tumour suppressor functions[2–5].

CYLD is a deubiquitinating enzyme that selectively hydrolyses K63- and M1-linked ubiquitin chains but exhibits very little activity towards K48-linked ubiquitin chains[6]. CYLD was shown to negatively regulate NF-κB and MAPK activation by removing ubiquitin chains from key signalling molecules including NEMO, TRAF2, TRAF6 and RIPK1 (refs 7–10). In addition, CYLD was proposed to inhibit Wnt signalling by deubiquitinating dishevelled[11]. Furthermore, CYLD was identified as an important regulator of TNF-induced apoptosis[12] and programmed necrosis[13], presumably by deubiquitinating RIPK1 to allow the formation of death-inducing protein complexes. Moreover, we showed previously that CYLD catalytic activity is important for the induction of RIPK3-mediated necroptosis *in vivo*[14,15].

CYLD-deficient mice did not develop spontaneous tumours but showed increased susceptibility in different models of carcinogenesis. Mice lacking CYLD exhibited exacerbated colon carcinogenesis induced by administration of the chemical carcinogen azoxymethane (AOM) followed by repeated cycles of dextran sulphate sodium (DSS) -induced inflammation[16]. In this study, the authors proposed that CYLD deficiency leads to increased NF-κB activation in macrophages resulting in elevated inflammation, which drives enhanced colon carcinogenesis. Moreover, CYLD deficiency resulted in chronic production of tumour-promoting cytokines by tumour-associated macrophages leading to more aggressive tumour growth in a syngeneic model of lung cancer[17]. CYLD-deficient animals also showed increased susceptibility to skin tumorigenesis induced by a single topical application of the chemical carcinogen 7,12-dimethylbenz(a)anthracene (DMBA) followed by repeated cycles of inflammation induced by application of 12-O-tetradecanoylphorbol-13-acetate (TPA) (ref. 18). In this study, Massoumi *et al.* suggested that CYLD deficiency in keratinocytes caused sustained ubiquitination and nuclear translocation of BCL3 in response to ultraviolet light or TPA, which induced elevated expression of Cyclin D1 resulting in increased keratinocyte proliferation and tumour growth. Furthermore, transgenic expression of catalytically inactive mutant CYLD in the epidermis also sensitized mice to DMBA/TPA-induced skin tumorigenesis[19]. However, in this study the authors suggested that increased activation of JNK and not NF-κB was responsible for the enhanced tumorigenesis caused by the expression of catalytically inactive CYLD. Therefore, it appears that CYLD may suppress tumour development by different mechanisms acting either in a cell intrinsic manner in premalignant epithelial cells or by regulating the tumour microenvironment by acting in myeloid cells.

The p53 transcription factor is a key tumour suppressor that is mutated in more than 50% of human cancers. p53 not only maintains genomic stability after cellular stress by controlling the expression of genes regulating cellular senescence, cell cycle progression, cell death and DNA repair but has also been recently implicated in the regulation of cellular metabolism, stem cell maintenance and the tumour microenvironment[20–22]. Regulation of p53 protein stability by the ubiquitin/proteasome system is the main mechanism controlling p53 function[23,24]. p53 ubiquitination by Mdm2 and a number of other E3 ubiquitin ligases including COP1, Pirh2, ARF-BP1, MSL2 and Parc mediates the degradation and controls the subsecular localization of p53 (refs 23,24). In response to cellular stress, HAUSP and a number of other deubiquitinating enzymes, including Otub1, USP10, USP29 and USP42 not only remove ubiquitin chains from p53 but also other proteins regulating p53 ubiquitination including Mdm2, inducing p53 stabilization (refs 23,24).

Here we show that CYLD acts in intestinal and skin epithelial cells to suppress DNA damage-induced tumour development and that this tumour suppressor function of CYLD is mediated at least in part by the regulation of p53 activation. Lack of CYLD catalytic activity results in impaired stabilization of p53, reduced expression of p53 target genes as well as reduced apoptosis in epithelial cells in response to genotoxic stress. Elevated transgenic expression of p53 restores DNA damage-induced cellular responses and partly normalizes the increased tumour susceptibility in mice expressing catalytically inactive CYLD *in vivo*. In addition, we show that CYLD regulates DNA damage-induced p53 responses in *Caenorhabditis elegans* suggesting that this function of CYLD is evolutionarily conserved. Mechanistically, we show that CYLD interacts with and deubiquitinates p53 in response to DNA damage. Moreover, we provide evidence that CYLD removes K48-linked ubiquitin chains from p53 indirectly by cleaving K63 chains, suggesting that p53 is decorated with complex ubiquitin chains.

## Results

**Epithelial CYLD suppresses AOM/DSS-induced colon cancer.** To study the function of CYLD deubiquitinase (DUB) activity in tumorigenesis we employed a conditional knockin mouse model allowing the cell-specific expression of the catalytically inactive CYLDR932X mutant (CYLDΔ932) (ref. 15). The R932X mutation in mouse CYLD is equivalent to the R936X mutation in human CYLD, which truncates subdomain III of the Histidine box that is essential for CYLD catalytic activity[8] and was found to cause cylindromas[1] (Supplementary Fig. 1a). By crossing mice carrying CYLDΔ932 floxed (CYLDΔ932$^{FL}$) alleles with LysM-Cre and Villin-Cre transgenics we generated two mouse lines expressing the catalytically inactive CYLDΔ932 mutant specifically in myeloid (CYLDΔ932$^{mye}$) or intestinal epithelial cells (IECs) (CYLDΔ932$^{IEC}$), respectively (Supplementary Fig. 1b–d). To address the myeloid cell-specific role of CYLD in inflammation-associated colon cancer, we induced colon carcinogenesis in CYLDΔ932$^{mye}$ mice by a single injection of the alkylating agent AOM followed by repeated cycles of inflammation induced by oral administration of DSS[25] (Fig. 1a). These experiments revealed that CYLDΔ932$^{mye}$ mice showed similar levels of inflammation and tumorigenesis compared with their CYLDΔ932$^{FL}$ littermates after AOM/DSS treatment (Fig. 1b–d). Therefore, in contrast to the suggestion by Zhang *et al.*[16] that CYLD deficiency in macrophages was responsible for the increased AOM/DSS-induced colon carcinogenesis in *Cyld*$^{-/-}$ mice to AOM/DSS-induced colon cancer, loss of CYLD catalytic activity specifically in myeloid cells did not sensitize mice to AOM/DSS-induced inflammation and tumour development.

We then assessed the role of epithelial CYLD by exposing CYLDΔ932$^{IEC}$ mice to AOM/DSS colon carcinogenesis. CYLDΔ932$^{IEC}$ mice reacted very strongly to AOM/DSS treatment, requiring the use of a milder protocol including only one DSS cycle to ensure survival of the mice (Fig. 1e). Under this protocol, CYLDΔ932$^{IEC}$ mice developed pronounced weight loss associated with severe colitis, while their CYLDΔ932$^{FL}$ littermates exhibited only mild colitis (Fig. 1f,g). Tumour evaluation on day 60 revealed that CYLDΔ932$^{IEC}$ mice harboured increased numbers of tumours in their colons compared with their CYLDΔ932$^{FL}$ littermates (Fig. 1h). Moreover, tumours in CYLDΔ932$^{IEC}$ mice were larger and more advanced with

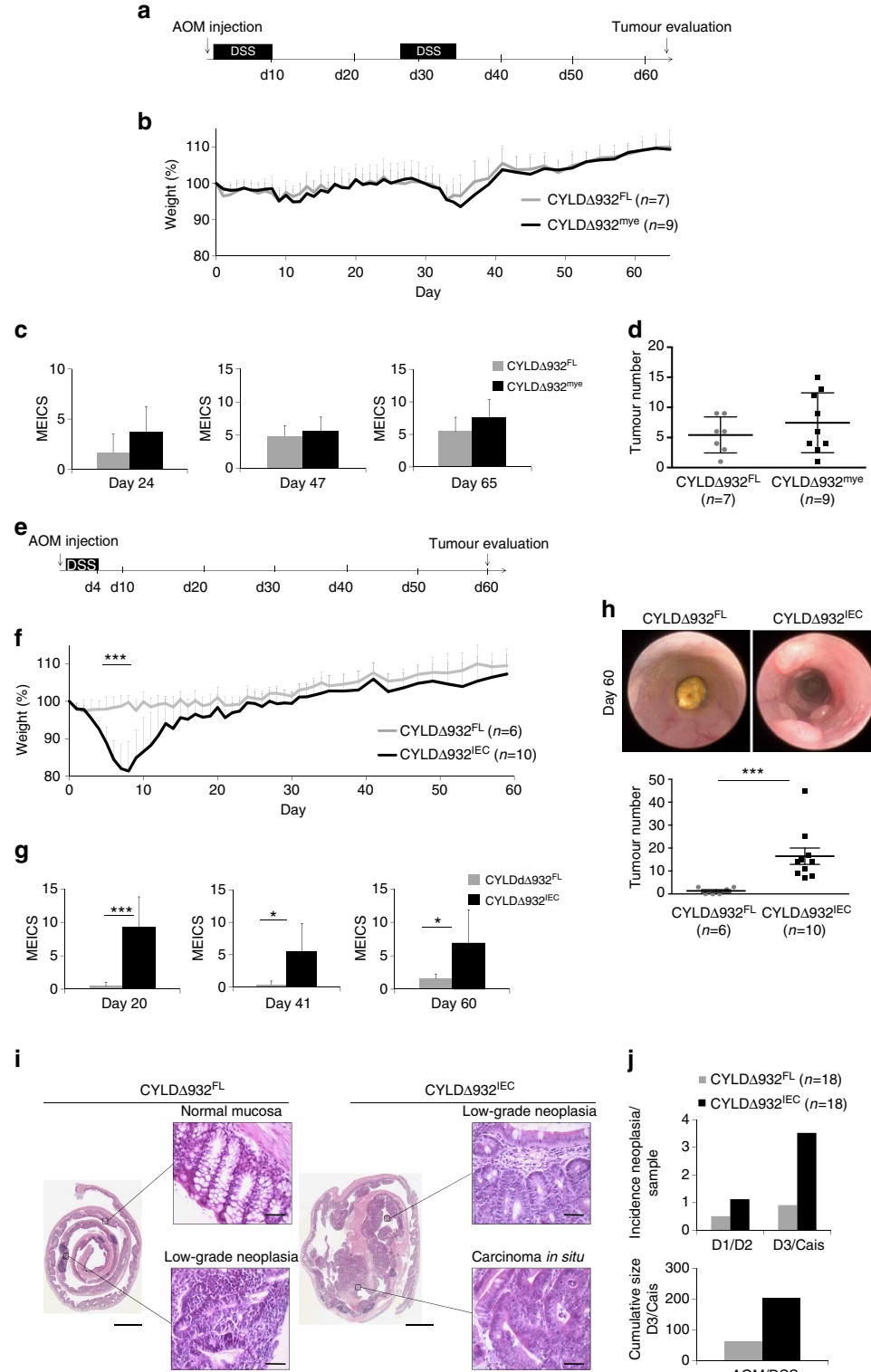

**Figure 1 | Epithelial CYLD suppresses AOM/DSS-induced colon tumorigenesis.** (**a**) CYLDΔ932[mye] and CYLDΔ932[FL] male littermates were injected with 10 mg kg[−1] AOM followed by two cycles of treatment with 2% DSS. (**b**) Graph showing body weight. (**c**) Quantification of murine endoscopic index of colitis severity (MEICS) on the indicated days. (**d**) Graph showing colon tumour numbers on day 65. One representative out of two independent experiments is shown. (**e**) CYLDΔ932[IEC] and CYLDΔ932[FL] male littermates were injected with 7.5 mg kg[−1] AOM followed by one cycle of treatment with 2% DSS. (**f**) Graph showing body weight. (**g**) MEICS on the indicated days. (**h**) Representative endoscopy images and graph showing colon tumour numbers. One representative out of five independent experiments is shown. (**i**) Representative Hematoxylin and Eosin (H&E) stained distal colon sections on day 60 from the indicated AOM/DSS-treated mice. Scale bars, 2,500 and 50 μm. (**j**) Graphs showing the incidence of low-grade intraepithelial neoplasia (D1/D2) and high-grade intraepithelial neoplasia/carcinoma in situ (D3/Cais), and the cumulative size of high-grade intraepithelial neoplasia/carcinoma in situ D3/(Cais) in the indicated AOM/DSS-exposed mouse lines. The evaluated samples comprise specimens from experiments shown in Fig. 1e–h and Supplementary Fig. 2a–d. For all experiments, data are shown as mean ± s.d. Statistical significance was determined with Student's t-test; *P ≤ 0.05, ***P ≤ 0.0005.

increased incidence of high-grade intraepithelial neoplasia and carcinoma *in situ* compared with CYLDΔ932[FL] littermates (Fig. 1i,j). Collectively, these results showed that CYLD catalytic activity functions in epithelial cells to suppress AOM/DSS-induced colon tumorigenesis.

**CYLD suppresses DNA damage-induced tumorigenesis**. To dissect whether CYLD suppresses colon tumorigenesis by preventing AOM-mediated tumour initiation or DSS-induced tumour promotion, we first studied the response of CYLDΔ932[IEC] mice to DSS (Fig. 2a). When treated with multiple cycles of DSS in the absence of AOM, CYLDΔ932[IEC] mice showed similar weight loss and colitis development compared with their CYLDΔ932[FL] littermates, while none of the mice showed colon tumours when sacrificed on day 180 (Fig. 2b,c). These results showed that epithelial CYLD does not regulate DSS-induced colon injury and inflammation, and suggested that CYLD catalytic activity might be important for the response of epithelial cells to AOM-induced DNA damage.

To assess the role of epithelial CYLD in DNA damage-driven tumorigenesis, we exposed CYLDΔ932[IEC] mice to a model of inflammation-independent intestinal tumour development induced by repeated injections of AOM (Fig. 2d). Endoscopic analysis on week 12 revealed the presence of colon tumours in all CYLDΔ932[IEC] mice, while none of their CYLDΔ932[FL] littermates showed tumours at this stage (Fig. 2e). Macroscopic and histological analysis of colons on week 18 revealed strongly increased AOM-induced tumorigenesis in CYLDΔ932[IEC] mice compared with CYLDΔ932[FL] littermates. CYLDΔ932[IEC] mice showed increased tumour incidence and developed more tumours, which were larger and more advanced with increased incidence of high-grade intraepithelial neoplasia and carcinoma *in situ* compared with CYLDΔ932[FL] animals (Fig. 2f–h). Thus, epithelial specific inhibition of CYLD catalytic activity sensitized mice to AOM-induced colon carcinogenesis.

To test whether CYLD regulates carcinogen-induced tumour formation in other epithelia, we generated mice expressing the catalytically inactive CYLDΔ932 mutant in epidermal keratinocytes (CYLDΔ932[epi]) by crossing CYLDΔ932[FL] mice with K14-Cre transgenics (Supplementary Fig. 1e,f). To induce skin tumours, we treated CYLDΔ932[epi] and CYLDΔ932[FL] littermates with eight consecutive weekly topical applications of the carcinogen DMBA on the shaved back skin (Fig. 2i). Evaluation of skin tumour development 12 weeks after the last DMBA application revealed the presence of papillomas in ~70% of the CYLDΔ932[epi] mice compared with ~30% of their littermate controls. Moreover, about two-thirds of the papillomas found in CYLDΔ932[epi] mice were classified as medium to big, while nearly all papillomas found in CYLDΔ932[FL] mice were small (Fig. 2j,k). Collectively, these results showed that CYLD catalytic activity in epithelial cells suppresses DNA damage-induced tumorigenesis in the intestine and skin.

**CYLD promotes DNA damage-induced apoptosis**. Cell death amid irreparable genomic lesions is a mechanism preventing DNA damage-driven tumour development. We reasoned that CYLD might prevent chemical carcinogen-induced tumour development by promoting DNA damage-induced cell death. Indeed, we found that 8 h after AOM injection colonic crypts in CYLDΔ932[IEC] mice contained reduced numbers of TUNEL-positive epithelial cells compared with CYLDΔ932[FL] littermates, indicating that loss of CYLD DUB activity protected IECs from DNA damage-induced death (Fig. 3a). CYLD was previously identified as an important regulator of cell death induced by RIP kinase 1 (RIPK1) downstream of death receptor and Toll-like

receptor signalling[12,13]. Furthermore, we showed that CYLD catalytic activity regulates RIPK3-induced necrosis in intestinal and skin epithelial cells *in vivo*[14,15]. Since RIPK1 was recently shown to regulate cellular responses to DNA damage[26], we hypothesized that CYLD might control DNA damage-induced cell death and tumour development by regulating RIP kinase-dependent pathways and tested this hypothesis using relevant genetic mouse models. We found that neither *Ripk3*[−/−] nor double FADD[IEC-KO]/*Ripk3*[−/−] mice, which lack RIPK3 in all cells and FADD specifically in IECs[15], showed increased susceptibility to AOM/DSS- or AOM-mediated intestinal tumorigenesis (Supplementary Fig. 2). These results suggest that the tumour suppressing role of CYLD is independent of RIPK3 and FADD/caspase-8 mediated cell death pathways.

**CYLD facilitates DNA damage-induced p53 activation**. Since AOM and DMBA are genotoxic agents, we reasoned that CYLD activity might increase DNA damage-induced cell death and therefore suppress tumour development by regulating the DNA damage response in epithelial cells. DNA damage induces the stabilization and activation of the transcription factor p53, which promotes cell cycle arrest to facilitate DNA repair or induces the death of cells carrying irreparable DNA lesions. Moreover, AOM-induced death of IECs depends on p53 signalling[27]. To address whether CYLD controls the p53-dependent death of epithelial cells in response to DNA damage we employed intestinal organoid cultures. Organoids from CYLDΔ932[IEC], CYLDΔ932[FL], p53[IEC-KO] and p53[FL] mice were treated with Camptothecin (CpT), a topoisomerase inhibitor inducing replication fork stalling and the formation of DNA double-strand breaks[28] or Mitomycin C (MMC), a chemical carcinogen that, similarly to AOM, induces DNA alkylation and subsequent interstrand crosslink (ICL) formation[29], and cell death was assessed by microscopic evaluation as well as detection of caspase-3 cleavage by immunocytochemical, immunoblot and fluorescence-activated cell sorting (FACS) analyses. As expected, CpT and MMC treatment induced epithelial cell death in wild type but not in p53-deficient organoids (Fig. 3b,c and Supplementary Fig. 3a–b). Consistent with our *in vivo* results, CYLDΔ932 mutant organoids were largely protected from CpT- and MMC-induced epithelial cell death compared with CYLDΔ932[FL] organoids (Fig. 3b–d and Supplementary Fig. 3a–c). Therefore, lack of CYLD catalytic activity inhibited the p53-dependent death of IECs in response to DNA damage *in vivo* and *in vitro*.

To address how CYLD regulates DNA damage-induced p53 responses we first examined whether CYLD regulates p53 stabilization in response to genotoxic stress. We found that CYLDΔ932 organoids showed strongly impaired stabilization of p53 as well as reduced expression of the p53-dependent genes *Cdkn1a*, *Bax* and *gadd45* in response to CpT (Fig. 3e,f). To assess whether lack of CYLD DUB activity affected DNA damage-induced p53 responses in other epithelial cells, we analysed CpT-induced p53 activation in primary epidermal keratinocytes. Similarly to IECs, CYLDΔ932 mutant keratinocytes showed impaired DNA damage-induced p53 stabilization and p53-dependent gene expression compared with controls (Fig. 3g,h). Together, these experiments showed that CYLD DUB activity is required for efficient stabilization and activation of p53 and the induction of cell death in primary epithelial cells exposed to DNA damage.

**p53 overexpression reverts tumorigenesis in CYLDΔ932[IEC] mice**. Our results suggested that CYLD regulates DNA damage-induced responses primarily by facilitating the optimal

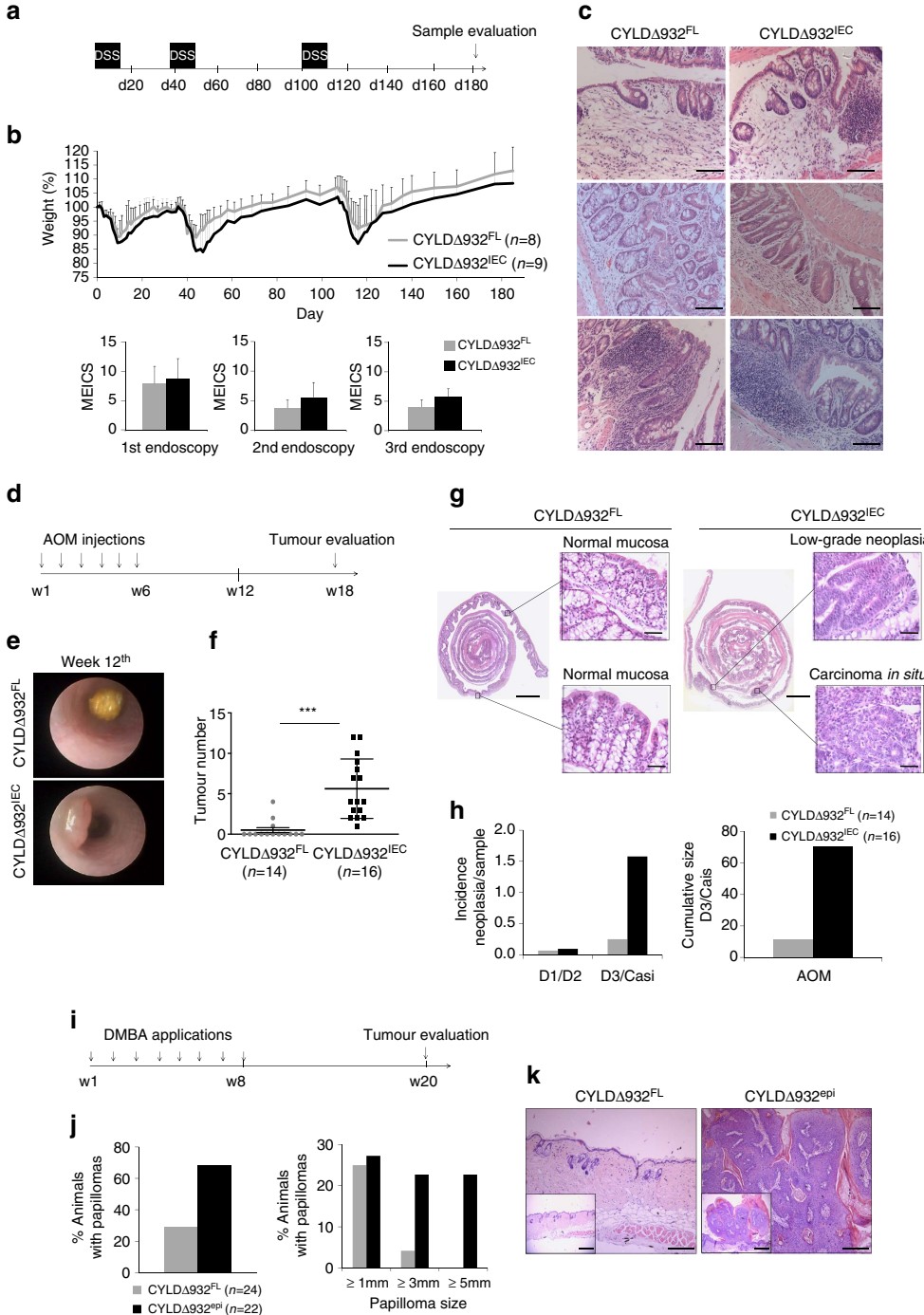

**Figure 2 | CYLD suppresses DNA damage-induced tumorigenesis independent of inflammation.** (**a**) CYLDΔ932[IEC] and CYLDΔ932[FL] male mice were given three 7-day cycles of 2% DSS in the drinking water. (**b**) Graphs showing body weight throughout the treatment and quantification of MEICS after each DSS cycle. (**c**) Representative H&E-stained distal colon sections from DSS-treated mice on day 180. Scale bar, 100 μm. One representative out of two independent experiments is shown. (**d**) CYLDΔ932[IEC] and CYLDΔ932[FL] female littermates received weekly injections of 10 mg kg$^{-1}$ AOM for 6 weeks. (**e**) Representative endoscopic images on week 12. (**f**) Graph showing colon tumour numbers on week 18. (**g**) Representative H&E-stained histological images on week 18. Scale bar, 2,500, 50 μm. (**h**) Graphs showing the incidence of low-grade intraepithelial neoplasia (D1/D2) and high-grade intraepithelial neoplasia/carcinoma *in situ* (D3/Cais), and the cumulative size of high-grade intraepithelial neoplasia/carcinoma *in situ* D3/(Cais) in the indicated mice exposed to AOM. The evaluated samples comprise specimens from experiments shown in Fig. 2d-h. Pooled data from three independent experiments is shown. (**i**) 25 μg of DMBA was applied on the back of CYLDΔ932[FL] and CYLDΔ932[epi] female mice once a week during 8 weeks. (**j**) Graphs showing quantification of papilloma development on week 20. (**k**) Representative H&E-stained histological images of skin from the indicated mice on week 20. Scale bar, 500 and 200 μm. Pooled data from three independent experiments are shown. For all experiments data are shown as mean ± s.d. Statistical significance was determined with Student's *t*-test; ***$P \leq 0.0005$.

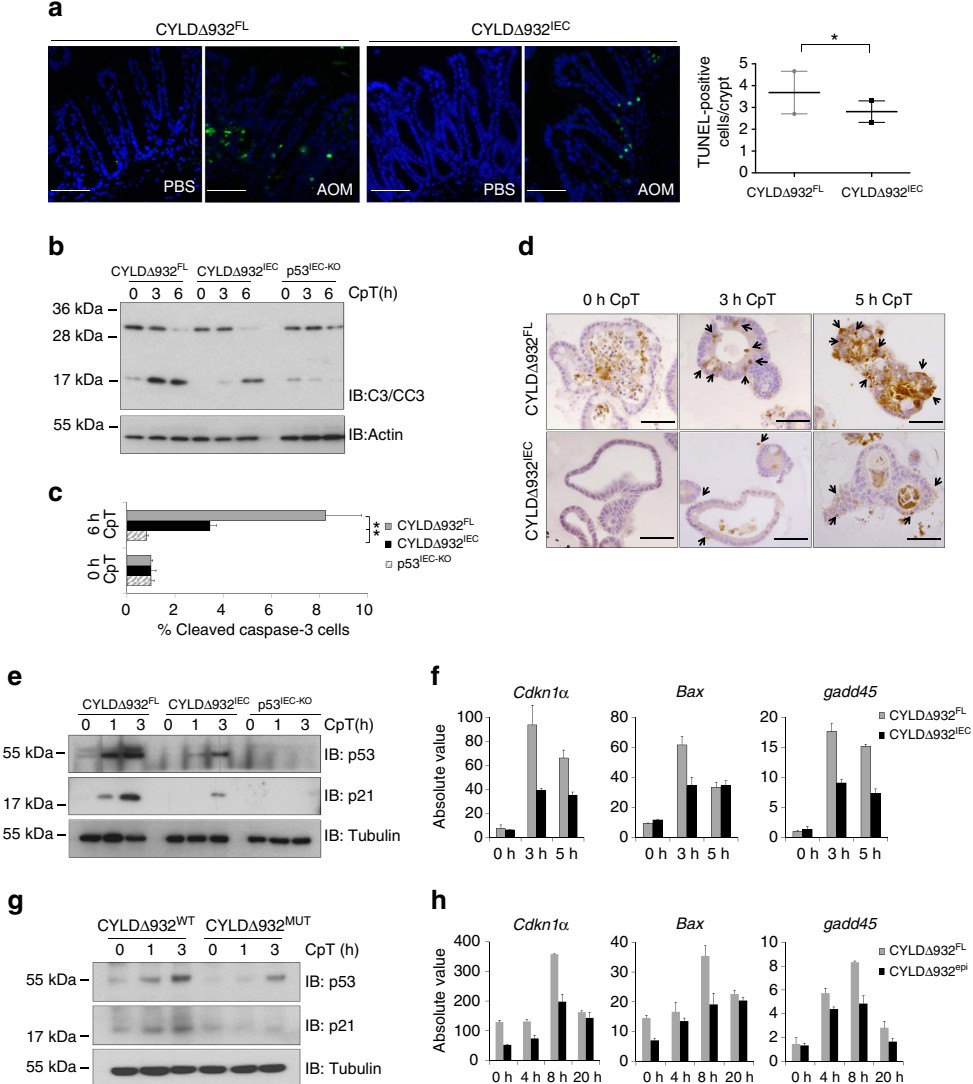

**Figure 3 | Impaired DNA damage-induced p53 activation in CYLDΔ932 mutant epithelial cells.** (**a**) Representative pictures from TUNEL-stained colon sections from the indicated mice 8 h after AOM or PBS injection and quantification of TUNEL-positive IECs in colonic crypts. Data shown as mean ± s.e.m. of 150 well-oriented crypts from three animals of each genotype. Statistical significance was determined with Student's t-test; *$P ≤ 0.05$. (**b**) Immunoblot analysis for caspase-3 (C3) and cleaved caspase-3 (CC3) in CpT-treated intestinal organoids prepared from the indicated mice. (**c**) FACS quantification of CpT-induced cleaved caspase-3-positive cells in organoids from the indicated mice. Mean ± s.d. of biological triplicates shown. Statistical significance was determined with Student's t-test **$P ≤ 0.005$. One representative out of three independent experiments is shown. (**d**) Immunohistochemical analysis for active caspase-3 in CpT-treated intestinal organoids from the indicated mice. Arrows indicate epithelial cells stained for active caspase 3. Note that the luminal contents of organoids show unspecific staining. Scale bar, 50 μm. (**e**–**h**) Immunoblot analysis of p53, p21 and tubulin and qRT-PCR analysis of p53 target gene expression in CpT-treated intestinal epithelial organoids (**e,f**) and primary epidermal keratinocytes (**g,h**) prepared from the indicated mouse lines. Graphs in **f** and **h** show means ± s.d. of the absolute values of technical duplicates, from one representative out of two independent experiments. qRT-PCR, quantitative real-time PCR.

stabilization of p53 in response to genotoxic stress. To assess whether increased expression of p53 could rescue impaired DNA damage responses in cells expressing mutant CYLD, we crossed the CYLDΔ932[IEC] mice with 'super p53' mice carrying supernumerary copies of the p53 gene in the form of large genomic transgenes[30]. Indeed, increased p53 expression could largely restore DNA damage-induced apoptosis and expression of p53 target genes in CYLDΔ932[IEC] intestinal organoids (Fig. 4a–c). In addition, CYLDΔ932[IEC] Superp53 mice developed less AOM/DSS- and AOM-induced colon tumours compared with CYLDΔ932[IEC] littermates, showing that increased p53 expression could partially restore the increased susceptibility of CYLDΔ932[IEC] mice to chemical carcinogen-induced colon

tumorigenesis (Fig. 4d,e). These results provide additional experimental evidence supporting that CYLD controls cellular responses to DNA damage by promoting the stabilization and activation of p53, and that regulation of p53-dependent responses to genotoxic stress contributes to the tumour suppressing function of CYLD.

**Inactivating CYLD mutations are found in several human cancers.** Mutations of the Cyld genomic locus have been previously reported in different tumour types[2,3,5]. We analysed sequencing data from 7,042 human tumour samples[31] and found that *CYLD* mutations were present in a number of different

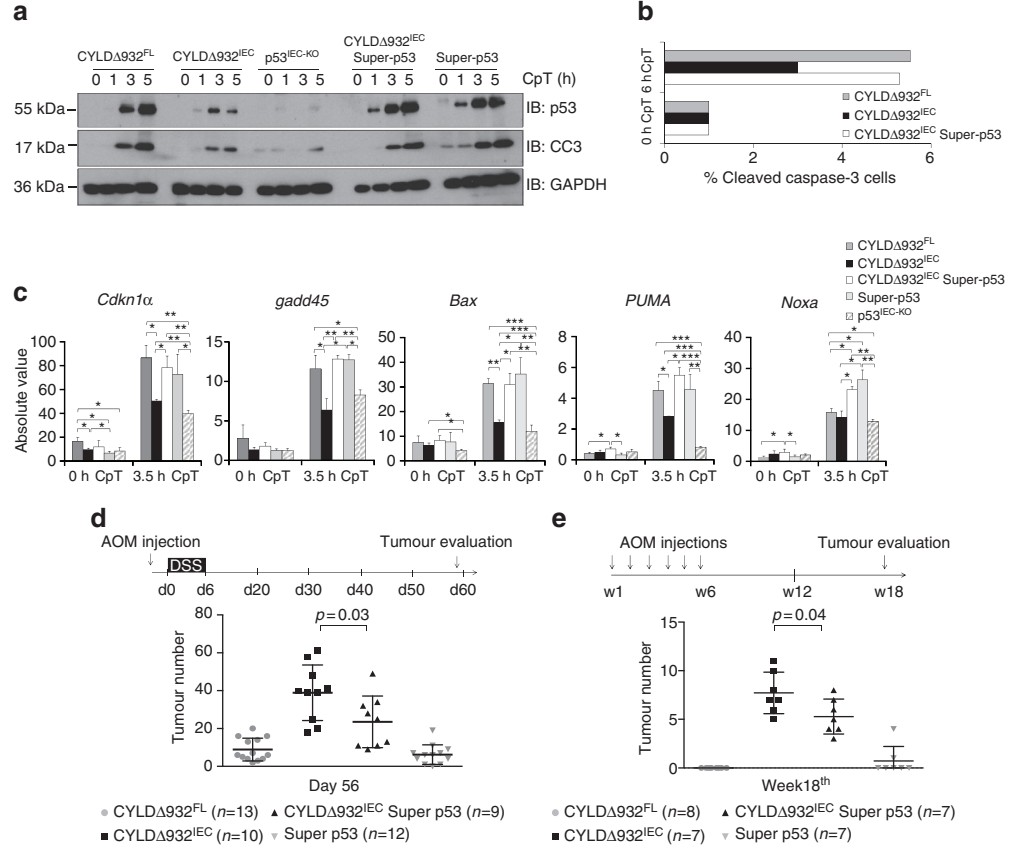

**Figure 4 | p53 overexpression restores DNA damage-induced cell death and reduces tumorigenesis in CYLDΔ932$^{IEC}$ mice.** Immunoblot analysis of p53, cleaved caspase-3 (CC3) and GAPDH (**a**), quantification of cleaved caspase-3-positive cells by FACS analysis (**b**) and qRT-PCR analysis of p53 target genes expression (**c**) in CpT-treated organoids from the indicated mice. Graphs in **c** show means of the absolute values ± s.d. of biological triplicates. Statistical significance was determined with Student's *t*-test; *$P \leq 0.05$, **$P \leq 0.005$, ***$P \leq 0.0005$. (**d**) Male littermate mice from the indicated mouse lines were injected with 7.5 mg kg$^{-1}$ AOM and 4 days later were given 2% DSS for 5 days. Graph shows colon tumour numbers on day 56. Pooled results from two different experiments are shown. (**e**) Female littermate mice from the indicated mice lines received weekly injections of 10 mg kg$^{-1}$ AOM for 6 weeks. Graph depicts colon tumour numbers on week 18. For the cancer experiment, data is shown as mean ± s.d. Statistical significance was determined with Student's *t*-test; *P* values are shown. qRT-PCR, quantitative real-time PCR.

cancers (Supplementary Fig. 4a). Determination of the frequency of inactivating mutations in 27,836 genes across cancer (COSMIC database) showed that inactivating mutations were significantly enriched in *CYLD* and that the *CYLD* mutation spectrum resembled the spectrum of known tumour suppressors (Supplementary Fig. 4b). In addition, we found a statistically significant association between the presence of CYLD mutation and the accumulation of mutations per tumour (Supplementary Fig. 4c), suggesting that loss of CYLD catalytic activity may favour the accumulation of somatic mutations in human cancers by impairing DNA damage-induced p53 responses. Moreover, CYLD was predicted to regulate p53 in an unbiased bioinformatics approach aiming to build a functional human protein interaction network by combining protein interaction, gene expression and gene ontology annotations with genome-wide cancer data sets[32], further supporting that regulation of p53 signalling by CYLD is functionally relevant for human cancer.

**CYLD regulates DNA damage-induced p53 responses in *C. elegans*.** To assess the functional conservation of the role of CYLD in regulating p53-dependent cellular responses to DNA damage, we investigated the function of the highly conserved *cyld-1* gene in *C. elegans*. Similar to mammals, nematodes activate

a conserved DNA damage checkpoint pathway that induces cell cycle arrest of mitotic germ cells and in meiotic pachytene cells activates the p53 homologue CEP-1 to trigger apoptosis[33–35] through transcriptional activation of the BH3 domain only proteins EGL-1 and CED-13 (ref. 36,37). In agreement with previous results[35], *cep-1* knockdown completely protected *C. elegans* from DNA damage-induced germ cell apoptosis (Fig. 5a). Strikingly, worms treated with RNA interference (RNAi)-mediated knockdown of *cyld-1* showed significantly reduced number of apoptotic cells following ionizing radiation (IR)-induced DNA damage (Fig. 5a). Likewise, a *cyld-1(tm3768)* mutant strain, which harbours a deletion of 496 bp in exon 14 that is predicted to result in the expression of truncated catalytically inactive CYLD-1 (Supplementary Fig. 5), showed reduced IR-induced germ cell death compared with wild-type worms (Fig. 5b).

We then examined whether increasing the levels of CEP-1/p53 could sensitize *cyld-1(tm3768)* worms to DNA damage-induced germ cell apoptosis in *cyld-1(tm3768)* mutant worms carrying the *gld-1*(op236) mutation. The Quaking homologue GLD-1 binds to the *cep-1/p53* mRNA and represses its translation. The *gld-1(op236)* mutant allele is defective in binding and repressing the *cep-1/p53* mRNA resulting in elevated CEP-1/p53 protein levels and increased DNA damage-induced apoptosis[38].

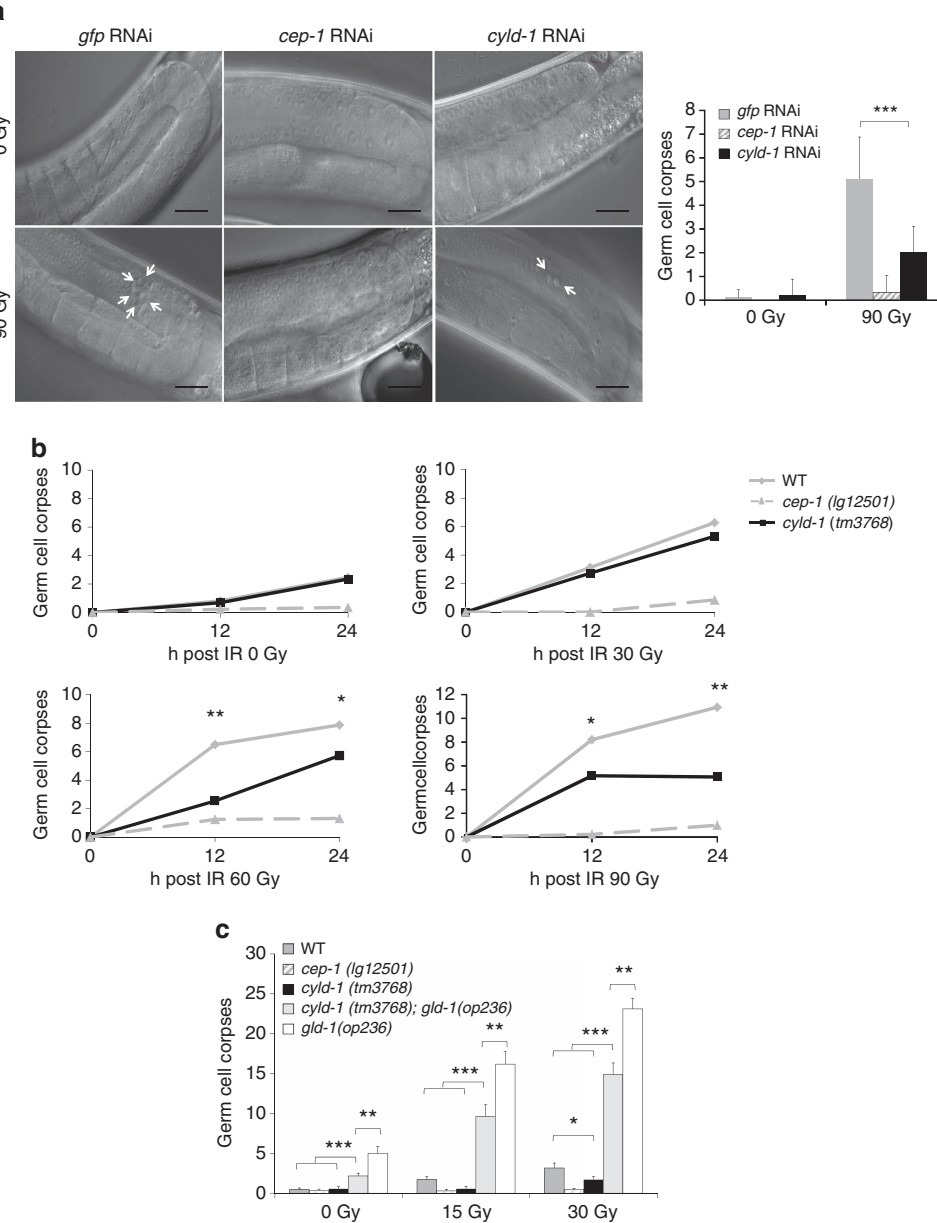

**Figure 5 | CYLD regulates CEP-1/p53-dependent cell death upon DNA damage in *C. elegans*.** (**a**) Representative pictures showing the pachytene region of the germline and quantification of corpses formation 24 h after 90 Gray (Gy) irradiation from *cep-1* and *cyld-1* RNAi fed worms (arrows indicate apoptotic corpses). Data shown as mean ± s.d is shown. Statistical significance was determined with Student's *t*-test; ***$P \leq 0.0005$. Scale bars, 20 µm. (**b**) Quantification of germ cell corpses in the indicated worm lines after irradiation. One representative out of three independent experiments is shown. Data shown as mean. Statistical significance was determined with Student's *t*-test; *$P \leq 0.05$, **$P \leq 0.005$. (**c**) Quantification of germ cell corpses in the indicated worms 24 h after irradiation. Data shown as mean ± s.d. is shown. Statistical significance was determined with Student's *t*-test; *$P \leq 0.05$, **$P \leq 0.005$, ***$P \leq 0.0005$. One representative out of three independent experiments is shown.

Indeed, *cyld-1(tm3768);gld-1(op236)* double-mutant worms displayed an intermediate phenotype between *cyld-1(tm3768)* and *gld-1(op236)* single mutants, demonstrating that increased levels of CEP-1/p53 partly restored the sensitivity of *cyld-1* mutant worms to DNA damage-induced apoptosis (Fig. 5c). In addition, these findings showed that loss of CYLD-1 catalytic activity partly inhibited the increased sensitivity of *gld-1(op236)* worms to apoptosis, providing additional evidence that CYLD-1 acts at the level of CEP-1/p53 to control the DNA damage response in *C. elegans*. Together, these results show that nematode CYLD-1 is required for the efficient activation of CEP-1/p53-dependent responses to DNA damage. Since the NF-κB signalling pathway is absent in *C. elegans*[39], these results also provide evidence that the role of CYLD in regulating DNA damage-induced p53 responses is not dependent on its function as a negative regulator of NF-κB activation. In agreement with this, lack of CYLD catalytic activity did not considerably affect the DNA damage-induced IκBα degradation and the transcription of NF-κB target genes in mouse primary epithelial cells, supporting an NF-κB-independent function of CYLD in the regulation of the DNA damage response (Supplementary Fig. 6).

**CYLD interacts with p53 in response to DNA damage.** Given the well-established function of CYLD as a deubiquitinating

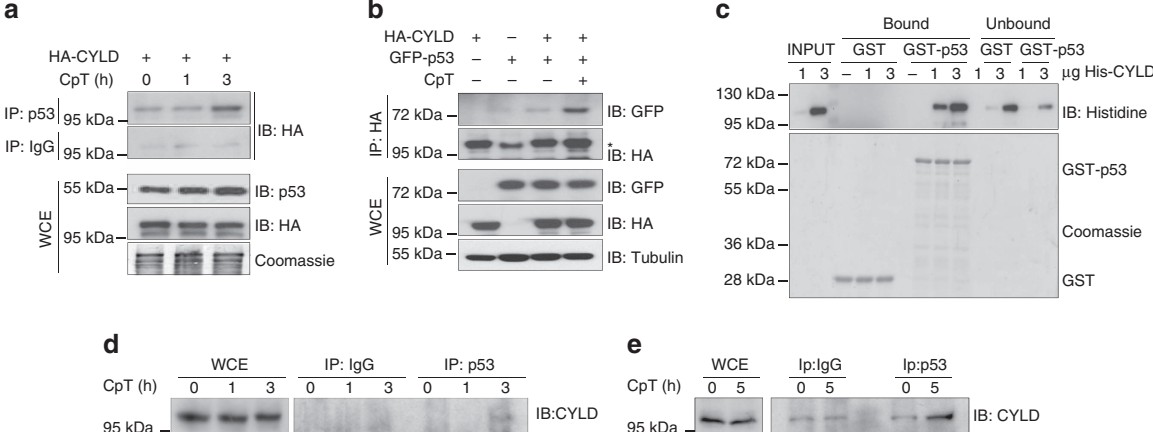

**Figure 6 | CYLD and p53 interact in response to DNA damage.** (**a**) p53 immunoprecipitation from CpT-treated HEK-293 T cells transfected with the indicated plasmids. Immunoblot with anti-HA antibodies shows co-immunoprecipitated HA-CYLD. Expression levels of p53 and HA-CYLD in whole cell extracts (WCE) are shown. (**b**) HA-CYLD immunoprecipitation from CpT-treated HEK-293 T cells transfected with the indicated plasmids. Immunoblot with anti-GFP antibodies shows co-immunoprecipitated GFP-P53. Expression levels of GFP-p53 and HA-CYLD in WCE are shown. *corresponds to an IgG nonspecific band. (**c**) GST pull-down assay showing direct interaction between GST-p53 and His-CYLD. Coomassie staining shows equal amount of GST and GST-p53 proteins. Endogenous p53 was immunoprecipitated from primary keratinocytes (**d**) and HCT116 cells (**e**) treated with CpT for the indicated time points. Immunoblot with anti-CYLD antibodies shows the co-immunoprecipitated CYLD. Endogenous CYLD levels in WCE are shown.

enzyme, we hypothesized that CYLD could regulate p53 activation by deubiquitination. We first tested whether CYLD and p53 interact with each other in cells exposed to DNA damage. We found that HA-CYLD expressed in HEK-293 T cells co-immunoprecipitated with endogenous p53 or overexpressed GFP-p53 in reciprocal IPs and that this interaction was enhanced after DNA damage (Fig. 6a,b). Furthermore, GST pull-down assays showed that recombinant His-CYLD bound recombinant GST-p53 but not GST, suggesting that CYLD directly interacts with p53 (Fig. 6c). In addition, endogenous CYLD co-immunoprecipitated with endogenous p53 in primary epidermal keratinocytes and HCT116 cells in response to DNA damage (Fig. 6d,e). Together, these results showed that CYLD interacts with p53 and this interaction is enhanced by genotoxic stress.

**CYLD deubiquitinates p53 facilitating its stabilization.** To test whether CYLD expression could alter p53 ubiquitination, we first used HEK-293 T cells. Expression of p53 together with HA-Ub in HEK-293 T cells resulted in robust ubiquitination of p53 also in the absence of DNA damage (Fig. 7a). Consistent with a role of CYLD in negatively regulating p53 ubiquitination, co-expression of HA-CYLD strongly reduced the ubiquitination of p53 (Fig. 7a and Supplementary Fig. 7a,b). In addition, overexpression of HA-CYLD diminished the ubiquitination and increased the stabilization of endogenous p53 in CpT-treated HCT116 cells (Fig. 7b,c). To assess whether CYLD DUB activity is required for the removal of ubiquitin chains from p53, we examined whether expression of two catalytically inactive CYLD mutants (the C-terminal truncated mutant R936X that is the human equivalent to the mouse R932X mutation and the CYLDH871N mutant where the putative catalytic Histidine is changed to Arginine abolishing DUB activity[8]) affected p53 ubiquitination. As shown in Fig. 7d and Supplementary Fig. 7c,d, in contrast to WT CYLD, the catalytically inactive CYLDR936X (R/X) and CYLDH871N (H/N) mutants did not diminish p53 ubiquitination although they bound p53, demonstrating that CYLD DUB activity is required to reduce p53 ubiquitination. Furthermore, recombinant His-CYLD reduced ubiquitination of Flag-p53 immunoprecipitated from CpT-treated HEK293T cells, showing that CYLD

can directly deubiquitinate p53 in a cell-free *in vitro* assay (Supplementary Fig.7e). Together, these results suggested that CYLD directly interacts with and deubiquitinates p53 facilitating its optimal stabilization in response to DNA damage.

**CYLD removes complex ubiquitin chains from p53.** It is generally believed that ubiquitination of p53 with K48-linked chains controls its constant degradation in unstressed cells, while inhibition of the formation as well as DUB-mediated removal of K48 chains is essential for p53 stabilization in response to DNA damage[24]. CYLD selectively hydrolyses K63- and M1-linked ubiquitin chains but exhibits very little activity towards K48-linked ubiquitin chains[6]. We therefore wondered whether CYLD could remove K48-linked ubiquitin chains from p53. To address this question, we first assessed the capacity of CYLD to reduce p53 ubiquitination in cells overexpressing HA-tagged Lys-to-Arg ubiquitin mutants that can only form K48- or K63-linked chains. This experiment showed that CYLD diminishes p53 ubiquitination in cells expressing HA-mutant ubiquitin forming K63 only chains, but also in cells expressing HA-mutant ubiquitin forming K48 only chains (Fig. 7e). As an alternative approach, we used K63 or K48 linkage-specific antibodies to assess the type of chains CYLD removes from p53. As shown in Fig. 7f, wild type but not catalytically inactive CYLD diminished both K63- and K48-linked ubiquitin chains from p53 in cells overexpressing wild-type ubiquitin (Fig. 7f). Together, these results provided evidence that CYLD removes also K48-linked ubiquitin chains from p53.

To understand how CYLD, a DUB that efficiently hydrolyses K63 and linear chains but shows little activity against K48-linked ubiquitin chains, removes K48 chains from p53 we performed ubiquitin chain restriction (UbiCRest) analysis[40] on ubiquitinated Flag-p53 (Flag-p53(Ub)n) immunoprecipitated from MG132/CpT-treated HEK-293 T cells. This experiment showed that USP21 (unspecific DUB) completely removed ubiquitin chains from p53, while OTUB1 (K48 linkage-specific DUB) and AMSH (K63-linkage-specific DUB), but not OTULIN (M1-linkage-specific DUB), strongly reduced p53 ubiquitination (Fig. 7g), suggesting that p53 is ubiquitinated with K48 and K63 but not

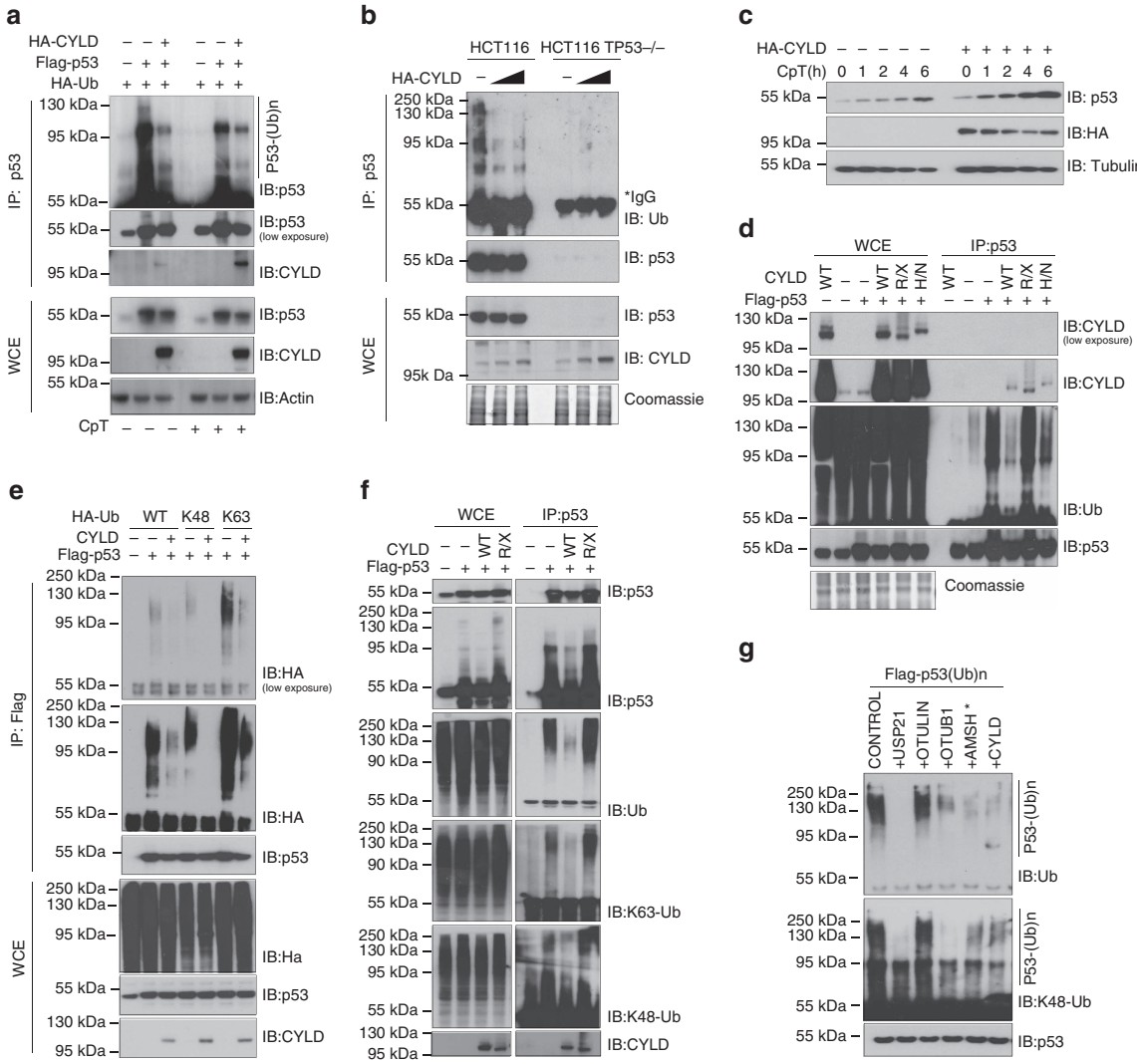

**Figure 7 | CYLD deubiquitinates p53 and induces its stabilization.** HEK-293 T (**a**,**d**) and HCT116 (**b**) cells were transfected with the indicated plasmids and were subsequently treated with MG-132/CpT. p53 was immunoprecipitated followed by immunoblot with anti-p53(DO-1), anti-ubiquitin and anti-CYLD antibodies as indicated. Expression levels of p53 and CYLD proteins are shown in immunoprecipitates and WCE. (**c**) HCT116 were transfected with HA-CYLD or an empty vector and treated with CpT for the indicated time points followed by immunoblot with anti-p53, HA and tubulin antibodies. (**e**,**f**) HEK-293 T cells were transfected with the indicated constructs and treated with MG-132/CpT. p53 was immunoprecipitated and its levels and type of ubiquitination were analysed by immunoblot using the indicated antibodies. (**g**) Ubiquitinated Flag-p53 (Flag-p53(Ub)n) was immunoprecipitated from MG-132/CpT-treated HEK-293 T cells and subjected to an *in vitro* deubiquitination assay using the indicated DUBs. Total and K48-linked ubiquitin chains on p53 are shown by immunoblot using specific antibodies.

linear chains. Interestingly, immunoblot analysis with K48 linkage-specific antibodies showed that USP21 and OTUB1 completely removed, but also AMSH strongly diminished K48-linked ubiquitin chains from p53 (Fig. 7g). Since AMSH is highly specific for K63 chains and cannot hydrolyse K48 linkages, its effect on removing K48 chains from p53 is most likely indirect, either by hydrolysing K63 linkages in a heterotypic K63/K48 ubiquitin chain background, or by cleaving intact K48 chains at the substrate-proximal ubiquitin. CYLD behaved similarly to AMSH in these experiments, suggesting that CYLD most likely also removes K48 chains from p53 indirectly.

## Discussion
Genomics studies in human cancers as well as experiments in mouse models have established CYLD as an important tumour suppressor in a variety of malignancies[1–5,16–19]. Most studies so

far attributed the tumour suppressing properties of CYLD to its function as negative regulator of NF-κB, MAPK and Wnt signalling. Nevertheless, the mechanisms of CYLD-mediated tumour suppression remain poorly understood. CYLD was recently shown to regulate death receptor-induced apoptosis and necroptosis, suggesting that regulation of death receptor-mediated programmed cell death could also contribute to its tumour suppressing functions. However, our *in vivo* genetic studies showing that double FADD/RIPK3 deficiency did not sensitize mice to AOM-induced colon tumorigenesis provided evidence that regulation of death receptor-induced programmed cell death is not critical for the tumour suppressing role of CYLD.

Our results presented here revealed a novel function of CYLD as a regulator of p53 signalling. The role of CYLD in regulating p53 is evolutionarily conserved as shown by our findings that CYLD deficiency impairs p53-dependent DNA damage-induced germ cell apoptosis in *C. elegans*. Since the role of ubiquitination

in the regulation of CEP-1/p53 signalling in *C. elegans* remains unclear, the precise mechanism by which CYLD controls DNA damage-induced CEP-1/p53 activation in worms remain to be investigated. Loss of CYLD catalytic activity reduced, but did not completely abolish, DNA damage-induced stabilization and activation of p53 in primary epithelial cells, suggesting that CYLD facilitates optimal p53 activation. This modulatory function of CYLD explains why CYLD-deficient mice do not phenocopy the severe phenotype of p53 knockout animals.

The stabilization and activation of p53 in response to cellular stress is regulated by the combined action of a number of E3 ubiquitin ligases and deubiquitinases[23,24]. p53 stability is controlled by K48 ubiquitin chains that target p53 for proteasome-dependent degradation. In addition, Ubc13 has been shown to elicit K63-dependent ubiquitination of p53 regulating its transcriptional activation[41]. Until recently, research on ubiquitin signalling has focused mainly on homogenous ubiquitin chains. However, there is increasing evidence from both *in vivo* and *in vitro* studies that single polyUb chains can contain multiple linkage types sequentially or in branched structures[42,43]. The properties of individual linkage types as well as their ability to be hydrolysed by DUBs remain intact when chains are branched[43,44]. Moreover, branched chains were shown to be processed faster by the 26S proteasome compared with homogeneous chains[42,44]. To our knowledge, the presence and possible functional role of complex ubiquitin chains on p53 has not been studied thus far. Our results using mutant ubiquitins and chain-specific ubiquitin antibodies showed that CYLD could remove not only K63 but also K48 chains from p53. Although we cannot exclude that CYLD could also hydrolyse directly K48 chains on p53, this is unlikely as previous experiments showed that CYLD efficiently hydrolyses K63 and linear chains but exhibits very little activity against K48 chains[45]. The results of the UbiCRest experiment showing that AMSH, a DUB that is highly specific for K63 chains could also remove K48 chains from p53 similarly to CYLD, suggest that these enzymes remove K48 chains indirectly. The most likely explanation of these findings is that p53 is decorated with mixed and/or branched K63/K48 ubiquitin chains and that CYLD can remove K48 chains indirectly by catalysing the cleavage of K63 linkages. We did not directly address the specific role of the proteasome in mediating CYLD-induced degradation of p53, as any experiment involving proteasome inhibition would result in p53 stabilization and thus would be inconclusive in proving the specific role of CYLD. However, given that CYLD mutation did not alter p53 mRNA levels (Supplementary Fig. 8), it is reasonable to conclude that CYLD stabilizes p53 by preventing its K48 chain-dependent proteasomal degradation. Collectively, these results provide a rational mechanism explaining how CYLD can regulate p53 protein stability directly, although it remains unclear whether CYLD targets additional proteins in the p53 pathway. Moreover, these findings provide experimental evidence that p53 is decorated with mixed and/or branched ubiquitin chains containing more than one linkages, suggesting that complex ubiquitin chains play an important role in regulating p53 signalling.

Our findings that CYLD catalytic activity is required for efficient DNA damage-induced activation of p53 suggest that regulation of p53 signalling is a major tumour suppressing mechanism of CYLD that is likely to synergize with its previously reported functions in negatively regulating NF-κB and Wnt signalling. By regulating these three pathways CYLD can affect three independent but interconnected processes that are central for tumorigenesis: genomic stability, inflammation and cell growth. The combined function of CYLD in modulating the activity of these pathways provides a rational mechanism for its important tumour suppressor role in a variety of human malignancies. In conclusion, our results reveal a novel tumour suppressor function of CYLD as a regulator of p53-dependent cellular responses to DNA damage.

## Methods

**Mice.** CYLDΔ932[FL] (ref. 15), *Rip3k*[−/−] (ref. 46), FADD[FL] (ref. 47), p53[FL] (ref. 48), Superp53 (ref. 30), Villin-Cre[49] and K14-Cre transgenic mice[50] have been previously described. For all experiments 8–10-week old gender-matched mice were used. Littermates carrying loxP-flanked alleles but not Cre served as control mice. Animals requiring veterinary attention were provided with appropriate care and excluded from the experiments. Mice were maintained at the animal facilities of the Institute for Genetics, University of Cologne, kept under a 12 h light cycle, and given a regular chow diet (Harlan, diet #2918) *ad libitum*. All animal procedures were conducted in accordance with European, national and institutional guidelines and protocols and were approved by local government authorities (Landesamt für Natur, Umwelt und Verbraucherschutz Nordrhein-Westfalen, Germany).

**Carcinogenesis protocols.** Two per cent DSS (MP, MW 36000-50000) was provided in drinking water *ad libitum*. AOM (Sigma) was dissolved in sterile PBS and injected intraperitoneally at a dose of 7.5 or 10 mg kg[−1] body weight. Colitis and tumour formation was monitored by high-resolution mini endoscopy. For DMBA-induced skin cancer, mice were treated topically on the shaved back skin with 25 μg of DMBA (Sigma) in 100 μl acetone once a week during 8 weeks. Animals were sacrificed after 12 weeks of latency. The tumour size scoring was done as follows; small: ≥1 mm; medium: ≥3 mm; and big: ≥5 mm diameter.

**High-resolution mini endoscopy.** Mice were anaesthetized using intraperitoneal injection of ketamine (Ratiopharm)/Rompun (Bayer) and a high-resolution mini-endoscope, denoted *Coloview* (Karl-Storz, Tuttlingen, Germany), was used to determine the murine endoscopic index of colitis severity as described previously[51].

**Histology and immunohistochemistry.** Samples were fixed overnight in 4% paraformaldehyde, embedded in paraffin and cut in 4 μm sections. Paraffin sections were rehydrated and heat-induced antigen retrieval was performed in 10 mM Sodium Citrate, 0.05% Tween-20 pH 6. Primary antibody used for IHC was anti-active Caspase 3 (Cell Signalling, 9661 (1:1,000 dilution). Biotinylated secondary antibodies were purchased from Perkin Elmer and Dako. Stainings were visualized with ABC Kit Vectastain Elite (Vector) and DAB substrate (DAKO). Incubation times with the DAB substrate were equal for all samples. General cell death was evaluated using DeadEnd Fluorometric TUNEL System (Promega). Pictures were taken with a fluorescence microscope (Leica) at the same exposure and intensity settings.

**Histopathological evaluation.** Intraepithelial neoplasias (IEN) were classified according to their grading in low- or high-grade/carcinoma *in situ*, this classification was made according to the WHO standard guidelines. The cumulative tumour size was calculated by summing up the size of all tumours found per sample. Tumour grade and size were evaluated blindly by two independent experienced pathologists. Histological sections that were not optimal for a proper evaluation were discarded from the analysis.

**Organoid cultures.** Mice were killed and intestines were flushed with PBS and cut longitudinally before villi were mechanically removed. Intestinal crypts were collected in PBS containing 2 mM EDTA. Crypts were plated in matrigel (BD) and supplemented with minimal growth medium containing ENR: 100 ng ml[−1] mNoggin, 100 ng ml[−1] R-Spondin, 50 ng ml[−1] mEGF. ENR was added to the medium every 3–4 days. Outgrowing crypts were passaged once a week. Organoids were stimulated in culture with CpT 10 μM (Sigma) or MMC 10 μg ml[−1] (Sigma) for the indicated times. For protein and RNA extraction, organoids were harvested with PBS, disrupted by pipetting and washed again with PBS in order to discard the organoid's lumen content.

**Keratinocyte isolation and culture.** Primary epidermis keratinocytes were isolated from the skin of mice between days 0 and 3 and cultured in minimal calcium medium (0.05 mM CaCl) supplemented with 4% Chelex-treated fetal calf serum and epidermal growth factor (10 ng ml[−1]). Keratinocytes were stimulated in culture with CpT 10 μM (Sigma) for the indicated times. Keratinocytes were lysed with RIPA lysis buffer containing proteases and phosphatase inhibitor tablets (Roche) for protein extraction and in Trizol reagent (Invitrogen) for RNA extraction.

**Cell transfection.** HEK (Life Technologies-Invitrogen), HCT116 and HCT116 TP53−/− (Horizon) negative for mycoplasma according to the MycoAlert Mycoplasma Detection Kit (Lonza) were cultured in DMEM with 10% FBS. Cells were plated at sub-confluence and transfected using lipofectamine 2000 (Life

Technologies) according to the manufacturer protocol. HCT116 and HCT116 TP53 −/− were validated by genotyping gDNA and cDNA by Horizon. HEK cells were validated by karyotyping by Invitrogen and employed for our expression vector experiments given their high capacity to be transfected and efficiency in exogenous gene expression.

**Plasmids.** Flag-P53 was provided by Dr Zhenkun Lou[52]; HA-CYLD were provided by Dr. Ana Bigas[53]; CYLD-WT, CYLDR936X and CYLDH871N mutant were provided by Dr. Gilles Courtois[8]; HA-Ubiquitin-WT (Addgene plasmid 17608), HA-Ubiquitin-K63-only (Addgene plasmid 17606) and HA-Ubiquitin-K48-only (Addgene plasmid 17605) were provided by Dr. Ted Dawson. GFP-p53 (Addgene plasmid 12091) was provided by Dr. T. Jacks. GST-p53 (Addgene plasmid 39479) was provided by Dr. Ie-Ming Shih. PCS2-Flag and pcDNA3-Ha plasmids were used to equalize amount of transfected DNA between samples.

**Immunoblotting.** Total protein cell extracts were separated by SDS–polyacrylamide gel electrophoresis gels and transferred to Immobilon-P polyvinylidene difluoride membranes (Millipore). Membranes were probed with primary antibodies; anti-p53(1C12) (Cell Signaling; 2524; 1:750 dilution), anti-p21(C-19) (Santa Cruz Biotechnology; sc-397; 1:1,000 dilution), anti-cleaved caspase-3 (Asp175) (Cell Signaling; 9661; 1:1,000 dilution), anti-phospho JNK (Invitrogen; 44-682G; 1:1,000 dilution), anti-JNK (Cell Signaling; 9252; 1:1,000 dilution), anti-IκBα (Santa cruz; sc-371; 1:1,000 dilution), anti-tubulin (Sigma; T6074; 1:5,000) and anti-actin (Santa Cruz Biotechnology; sc-1616; 1:1,000 dilution) antibodies at 4° O/N. Membranes were incubated with secondary horseradish peroxidase-coupled antibodies (GE Healthcare and Jackson Immune Research) and developed with chemiluminescent detection substrate (GE Healthcare and Thermo Scientific).

**Quantitative RT-PCR.** Total RNA was extracted with Trizol Reagent (Invitrogen) and RNeasy Columns (Qiagen) and cDNA was prepared with the SuperscriptIII cDNA-synthesis Kit (Invitrogen). PCR with reverse transcription was performed by TaqMan analysis (Applied Biosystems) using the following primers: Cdkn1α (Mm00432448_m1); Bax (Mm 00432051_m1); gadd45 (Mm 00432802_m1); PUMA (Mm 00519268_m1); Noxa (Mm 00451763_m1) and TATA-Box binding protein (Tbp) (Mm00446973_m1). Tbp was used as reference gene.

**Southern blotting.** Genomic DNA extraction, digestion and Southern blotting were performed according to standard protocols. The probe used for Southern blot analysis of the CYLD mutation was amplified using primers: sense: 5′-TCATGGCCAGCAGTCTCGAAG-3′; anti-sense: 5′-TTTCTGTGGGCCT ACATACGG-3′.

**Cell death assays.** Three days after splitting, intestinal organoids were treated with CpT (Sigma) 10 μM for the indicated time points. To get single-cell suspension, organoids were harvested with Tryple solution (Life Technologies), passed four times through a syringe (23G needle) and incubated at 37° for 5 min. Digestion was stopped with 2%FCS/PBS solution. Cells were then incubated with PBS containing amine-reactive dye (LIVE/DEAD dye, Live Technologies) washed, fixed with 2% PFA and permeabilized with PBS containing 1%BSA, 0.1%TritonX-100 in PBS. Cells were then stained with anti-cleaved caspase-3 (Asp175) antibodies (Cell Signaling; 9661; 1:50 dilution), followed by secondary antibodies coupled to ALEXA Fluor 488 (Molecular Probes; 1:200 dilution). Cleaved caspase 3 only positive cells were analysed using a FACS Calibur (BD).

**Ubiquitinated p53 immunoprecipitation and detection.** Forty-eight hours after transfection, cells were incubated sequentially with MG-132 (Calbiochem) 20 μM and CpT (Sigma) 10 μM for 7 and 3 h, respectively. Cell were lysed in IP buffer (8 M Urea, 20 mM Tris pH 7.5, 135 mM NaCl, 1% Triton X-100, 10% glycerol, 1.5 mM MgCl₂, 5 mM EDTA, 2 mM N-ethylmaleimide (NEM) plus proteases and phosphatase inhibitor tablets (Roche) and briefly sonicated. Urea was diluted to 1 M with urea-free IP buffer before antibody was added. IPs were done with anti p53(1C12) (Cell Signalling; 2524; 3 μg) antibodies or an isotype control IgGs (Santa Cruz Biothecnology; 3 μg) at 4° O/N. IgGs were captured with ProteinA-Magnetic Beads (Life technologies). For the FLAG IPs, anti-Flag M2 Magnetic Beads (Sigma; M8823; 3 μg) were used. Proteins were eluted with LDS reducing sample buffer (Bio-Rad) by heating at 70 °C for 3 min and were resolved using NuPAGE 4–12% Bis-Tris gels (Life Technologies) and transferred O/N at 20 V by wet transfer to nitrocellulose membranes. Membranes were probed with anti-p53 (DO-I) (Santa Cruz Biotechnologies; sc-126; 1:1000 dilution), anti-ubiquitin (Santa Cruz Biotechnologies; sc-8017; 1:1000 dilution), anti-ubiquitin K48-specific Apu-2 (Millipore; 05-1307; 1:1000 dilution), anti-ubiquitin K63-specific Apu-3 (Millipore; 05-1308; 1:1000 dilution), anti-Ha (12CA5) (Roche; 11 583 816 001; 1:1000 dilution) and anti-CYLD (provided by Dr R. Masoumi and Cell Signaling; 8462 S; 1:1000 dilution) antibodies at 4° O/N. Membranes were incubated with secondary horseradish peroxidase-coupled antibodies (GE Healthcare and Jackson ImmuneResearch) and developed with chemiluminescent detection substrate (GE Healthcare and Thermo Scientific). In all immunoprecipitation experiments the

whole cell extracts fraction correspond to 5% of the total lysate subjected to the immunoprecipitation, while 50% (for overexpressed proteins IPs) or 100% (for endogenous proteins IPs) of the total immunoprecipitation was loaded for immunoblot analysis.

**Pull-down assay.** GST fusion proteins were purified from the Escherichia coli BL-21 strain in lysis buffer using glutathione-Sepharose (Amersham Bioscience). Sepharose beads with GST fusion proteins where incubated with the indicated HEK-293 T-transfected cell lysate in binding buffer O/N at 4 °C. Beads were washed with binding buffer and proteins were eluted with LDS reducing sample buffer (Bio-Rad) by heating at 70 °C.

Lysis buffer: 20 mM Tris-HCl, pH 7.4, 1 mM NaCl, 0.2 mM EDTA, 1 mM dithiothreitol, 1 mg ml⁻¹ lysozyme, 1 mM PMSF and a protease inhibitor tablet (Roche).

Binding buffer: 25 mM Tris-HCl, pH 7.5, 200 mM NaCl, 0.2% Nonidet P-40, 1 mM DTT, 1 mM EDTA, 10% Glycerol and a proteases inhibitor tablet (Roche).

**Ubiquitin chain-restriction analysis on p53.** Ubiquitin chain restriction analysis was performed as previously described[40,54]. Briefly, ubiquitinated p53 was immunoprecipitated from transfected HEK-293 T cells, as described above, and incubated with the different deubiquitinases (DUBs) (USP21, 1.5 μM; OTULIN, 1 μM; OTUB1, 15 μM; AMSH*, 9 μM; CYLD, 0.7 μM) in deubiquitination reaction buffer (70 μM Tris (pH 7.5), 170 μM NaCl and 13 μM DTT) during 30 min at 37°. Beads were washed with deubiquitination reaction buffer and proteins were eluted with LDS reducing sample buffer (Bio-Rad) by heating at 70 °C for 3 min.

**C. elegans strains and growth conditions.** The N2 Bristol strain was used as wild type. The deletion mutant cep-1(lg12501) carries a 1213 bp deletion corresponding to 30,458–31,670 on cosmid F52B5 and takes out a large part of the cep-1 open reading frame. The mutant cyld-1(tm3768) harbours a deletion of 496 bp in the exon 14 of the gene. The gld-1(op236) mutant harbours a V to F substitution at amino acid 276 in the cep-1 mRNA binding domain. Mutant strains were backcrossed against N2 at least four times. Worms were grown under standard laboratory conditions (20 °C) on nematode growth medium (NGM) agar containing OP50 Escherichia coli.

**RNA interference treatment in C. elegans.** Overnight cultures of HT115 bacteria containing specific RNAi constructs were grown in lysogeny broth media containing ampicillin. RNAi expression was induced by adding 1 mM Isopropyl β-D-1-thiogalactopyranoside (IPTG) and incubating the cultures at 37 °C for 20 min before seeding the bacteria on NGM agar supplemented with ampicillin and 3 mM IPTG. Worms were used for experiments after 2 generations of RNAi feeding.

Cyld-1 RNAi sequence (Ahringer collection[55], chromosome III, clone F40F12, ORF F40F12.5): 5′-TCAGATAGTCCTTCTCGATCAGCCATCGAATCAAAGAA TACCCATTGATTCGATGAAGTTCGAACATAAGCAACGTAATGAGATGTT TCTATGCACAGAACCGCTGAGAGAACCATTTTGTGGGAGTGTGGTTTCT TTTGTGGTTTTCCAGGTGGGTAGAGATCTCGACTCTTGTGATCTTCAATT TCCGGCAAAAGATGTGTATGGTGGAAACATTTTCTGCAGAATATCACTT CCGAATAGAAAACTCTTCTCGTCAGGAAACACGTCGGGCAGTAGACTTC TGAGCATGCTTGACATTTTGAACAAGCCGGAACCGCTCCAGCAACGAAC GGAGTAATGTCAATTGTTTCCAGAGGAAGAATCTTATCAAACACTTTCTG TTGTCCATATCTTGGTAATTGCATAATCAATACTGGAGGAGCTTTTGCAA ATGTTACTTGAGCTGATCTCATATGCCGTTCGAGCAAATGCTGTGAAGTT GCTGCACCTCCCAGCCAATCATCCACCACAATTGGTACAAGATATTGAG AATCTTTTGCGTGATTTTGTCCGCTGAAAATTGAGAATATTAATACTAAT TGATTACTTTCGATATAAATAAGAATAAATTGGAAAAAAACTAACATCA ACTTTATGAATGGCTCAGCATGGAAAACTTTGGAGAATATGAATCCAAG AATCTCTTCTGGATCTTTTTCTTCATTTGTAAGCCCTGTTACATGCGGCA TGAGCTCAGCGAGCAATTTTCGAAGTTTCATGACATGATCCGCTCGCAC ATAATGAACTTTTCGAAGAGGGAACACGATCTCGTGGGCCAGAATTTTT TGAAATTGTTGAGCCGTTTCTGATCCTTTGATCGATTTTTCAAGAAGACT GAAATTATATTTTAAAATTTTTTATTGAGAGCATGCCAAACTTACAAATC AAAGCAAGTGGTTTGGACAAACATTGCATACAACGTGGCATCCAGATAA CACGAGTTACAGTAACCCTGAATTCCTTTCTGCCTGCCAACTAATTGTTG CATGTCCTTTGCAATTCCACATTTTTGCTTCTCCACTCCTGAATCCATACT TCCGAAATCTTCAGTTCGCCGGCTGATGTTATTATACGTTTGGCTAGTTG GATAGGTTGAA-3′.

cep-1 RNAi sequence (Ahringer collection[55], chromosome I, clone F52B5, ORF F52B5.5): 5′-AAAAATTCTAGGCCAGTCTGGAGTGCATTTAAAAACGCGGCAAT GTTTACCGCGTTTCAAATCAATTTTTTGCTAATTTTTTAATAACACTTATTGC TTGATCTTTATTCATAATCAAATACATTTGTGAATTGTTGATCTTTTTTTTG AACATTTGTTTTCACTCTTTGAATTGGTTCTTTTGAAAAAGCTCTTTATAATC GAGTAAAATTAATAGTGAAAATCTCTCTTTAAACATTTTAATATACTAAATA AAAACCATTTCAGAAACGAAATTCTCCACGCATACATCAAACAAGTTCGA ATTGTTGCCTATCCACGACGTGACTGGAAGAATTTCTGTGAGCGAGAAG ACGCAAAACAAAAGGATTTCAGATTTCCCGAGTTACCTGCCTACAAGAA GGCGAGCCTAGAATCGATAAATATCAAACAAGAGGTCAATCTAGAGAAC ATGTTCAACGTGACCAATACTACTGCACAGGTTTGTCTAGAATATGATTT AAAAGAATTTGGAAAATAGTAAATAACTTCAATTTCCAAAATCATGCCTTG

TAGAAATATCAATTTTCTGCCAGAAGAATATACCATATTTATCATGATTCT
TTTCAATTTTCTATAGAAAAACGTTTTCTTCTTGTTTCCGTCGAATGAACAA
AAACTGAATTGAAATTCGATATCTTATCTCCCAATGACTTCCTTTATGCGTA
ATGTTTGCTCGACCACAAAGCTTTGAAATGTTTTATTTCGATCTAATTTTTT
AAATATATATGCTCTAATTCCCAAGACATTCAATTATTTTCTCAACGCAATA
TGAAATAAGATTTTAAATTTACAGATGGAACCATCAACTTCATATTCATCT
CCATCAAACAGTAATAATCGGAAGAGATTTTTGAATGAGTGTGATTCTC
CAAATAATGATTATACAATGATGCACAGAACTCCACCAGTAACAGGTTA
TGCAAGTCGTCTTCATGGATGCGTTCCTCCGATTGAAACTGAACACGAA
AACTGTCAACTCCCGTCGATGAAGAGAAGTCGCTGTACCAATTATTCG
TTTAGAACGCTCACTGTAAGTATTTTTAGCTCACTCTTAAAGCAATATTAT
AATTAATTTTGATTCACAATAAATTTCAGCTGTCGACTGCTGAGTATACAA
AAGTCGTCGAATTTCTGGCACGCGAAGCAAAAGTTCCCAGATACACTT
GGGTTCCGACGCAAGTAGTCTCCCATATATTGCCAACTGAAGGACTTGA
AAGGTATTTATAAAGATTTATAAAGATTGCCGTAATTTCTTTTTTTTGAAGA
TTCCTCACCGCTATAAAAGCAGGGCACGATTCAGTGTTGTTCAATGCAA
ACGGAATTTATACAATGGGGGATATGATTAGAGAATTCGAGAAACATAA
TGACATCTTCGAAAGAATTGGTATCGATTCTTCGAAATTGTCGAAATAC
TACGAAGCGTTTCTCAGCTTTTACCGCATCCAGGAAGCGATGAAACTG
CCAAAGTAAAAATCATATCACCACCTGGTTTAATCGCCTAATTTGTTTTCA
CAA-3′.

**Irradiation experiments in C. elegans.** For ionizing radiation experiments, synchronized populations of L1 worms were obtained by passing mixed-stage cultures through 11-micrometre filter. Worms were then grown at 20 °C on Petri dishes containing NGM agar supplemented with OP50 E. coli. 48 h later late L4 worms were picked on new plates and irradiated using ISOVOLT, Titan E machine (GE) in combination with a 0.5 mm aluminium filter. Germ cell corpses were scored blindy as described[56]. Quantification of corpses formation upon irradiation is shown as mean of germ cell corpses/gonad ($n = 20–25$) ± s.d.

**Mutational analysis of human tumour samples.** To determine the CYLD mutation frequency across different cancer entities, sequencing files from 7,042 human tumour samples[31] (Sanger Institute Server) were analysed. We used the Human Genome HG19 annotation to link each mutation to its respective gene. Samples with low tumour content or weak sequencing criteria and mutations that were annotated as non-somatic were excluded from further analysis. A sample was classified as Cyld mutant, if it carried at least one somatic mutation in the coding region of the Cyld gene. We analysed primary tumour sites, for which at least two independent Cyld mutant samples were available (recurrent mutation).

To characterize the impact of CYLD mutations on the mutation spectrum size, we counted the number of mutations per sample between Cyld mutant and wild type and compared the average mutation count by using Student's t-test. For all primary tumour sites, which contained recurrent Cyld mutations, at least 175 samples were examined. Therefore samples were large enough to fulfil the prerequisites of Student's t-test as well as in order to estimate the variation within each group of data. In all groups compared by t-test variance was similar enough to apply a homoscedastic testing.

Further, we compared the mutation types in CYLD with the mutation spectra of known tumour suppressors and oncogenes. We determined the frequency of inactivating mutations in 27,836 genes across the entire COSMIC database. For each gene, we tested whether inactivating mutations were significantly enriched or underrepresented in its mutation spectrum, assuming a uniform distribution of inactivating mutations ($\chi^2$-test). Significance values were corrected for multiple testing (Benjamini–Hochberg correction).

**Statistical analysis.** For the animal experiments in order to determine group size necessary for adequate statistical power, power analysis employing the program G*power[57] was performed using preliminary data sets. Mice of the indicated genotype were assigned at random to groups. Mouse studies were performed in a blinded fashion. Results are shown as mean ± s.d. or ± s.e.m. Statistical significance was determined with the Student's t-test; $*P \leq 0.05$, $** P \leq 0.005$, $***P \leq 0.0005$. Groups were large enough to fulfil the prerequisites of Student's t-test as well as to determine that the variance between groups is similar.

**Reproducibility of experiments.** For the animal experiments; experiments presented in Figs 1b–d,f–h and 2b were repeated twice obtaining similar results. Results shown in Fig. 2f,j correspond to pooled data from 3 independent experiments. Results shown in Fig. 4d, Supplementary Fig. 2b–d and Supplementary Fig. 2f correspond to pooled data from two independent experiments. The histological pictures shown in Figs 1i and 2c,g,k are representative for 36, 17, 30, 46 animals in total, respectively.

Immunofluorescence images presented in Fig. 3a are representative for six animals in total. Microscopy images presented in Supplementary Fig. 3 are representative for two different experiments.

FACS quantification of cleaved caspase-3 experiments shown in Figs 3c and 4b were repeated three times with technical duplicates when possible, obtaining similar results.

qPCR analysis shown in Fig. 3f,h are representative from two independent repetitions using different primary cell isolations and technical duplicates. In qPCR

analysis in Fig. 4c biological triplicates were used. Results shown in Supplementary Fig. 6a are representative from two independent repetitions.

Western blot shown in Fig. 3b,e,g and Fig. 4a were repeated at least three times. Interaction experiments shown in Fig. 6a were repeated four times; and experiments shown in Fig. 6b–e were repeated twice obtaining similar results.

Ubiquitination experiments shown in Fig. 7a,d were repeated four times. Experiments presented in Fig. 7e,f were reproduced twice. The ubiquitin chain restriction analysis shown in Fig. 7g was reproduced three times obtaining comparable results.

For the C. elegans data; experiments shown in Fig. 5a were repeated twice and experiments presented in Fig. 5b,c were repeated three times obtaining comparable results.

**Data availability.** The mutational analysis of human tumour samples presented in Supplementary Fig. 4 was performed on the sequencing data reported by Alexandrov et al.[31], which is available from ftp://ftp.sanger.ac.uk/pub/cancer/AlexandrovEtAl. The authors declare that all other data supporting the findings of this study are available within the article and its Supplementary Information files or from the corresponding author upon reasonable request.

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

## Acknowledgements

*C. elegans* strains were kindly provided by the Mitani laboratory and the CGC (funded by the US National Institutes of Health (NIH) Office of Research Infrastructure Programs (P40 OD010440). RIPK3 knockout mice were kindly provided by V. Dixit (Genentech) and Superp53 mice by M. Serrano (CNIO). B.S. acknowledges funding from the DFG (CECAD, SFB 829 and KFO 286), the ERC (Starting grant 260383), Marie Curie (FP7 ITN CodeAge 316354, aDDRess 316390, MARRIAGE 316964 and ERG 239330), the German-Israeli Foundation (GIF, 2213-1935.13/2008 and 1104-68.11/2010), the Deutsche Krebshilfe (109453) and the BMBF (SyBaCol). R.K.T. is funded by the German Cancer Aid (Deutsche Krebshilfe, grant ID: 109679), by the German Ministry of Science and Education (BMBF) as part of the e:Med program (01ZX1303A), by the German federal state North Rhine Westphalia (NRW) and the European Union as part of the EFRE initiative (grant EFRE-0800397). D.K. is funded by the Medical Research Council [U105192732], the ERC [309756] and the Lister Institute for Preventive Medicine. M.P. acknowledges funding from the DFG (SFB670, SFB829 and SPP1656), the ERC (grant agreement no. 323040), the European Commission (FP7 grant 223151 (InflaCare)), the Deutsche Krebshilfe (Grant 110302), the Else Kröner-Fresenius-Stiftung and the Helmholtz Alliance (PCCC). V.F.-M. and M.A.E. were supported by EMBO long-term fellowships.

## Author contributions

P.-S.W and V.F.-M. performed the *in vivo* tumorigenesis experiments. V.F.-M. performed the biochemical analysis of DNA damage responses in mammalian cells. V.F.-M., M.A.E. and M.S. performed the *C. elegans* experiments. A.A and R.B. performed histological analysis of colon tumours. B.S. supervised the *C. elegans* experiments. F.D. and R.K.T. performed the mutational analysis of human tumour samples. D.K. provided reagents and advice for the characterization of p53 ubiquitination and UbiCRest analysis. M.P. coordinated the project and together with V.F.-M. designed the study and wrote the paper.

## Additional information

**Competing financial interests:** R.K.T. is a co-founder and shareholder of Blackfield AG and New Oncology. R.K.T. has received research grants from AstraZeneca, EOS and Merck KgaA, and has received honoraria from AstraZeneca, Bayer, Blackfield AG/New Oncology, Boehringer Ingelheim, Clovis Oncology, Daiichi-Sankyo, Eli Lilly, Johnson & Johnson, Merck KgaA, MSD, Puma, Roche and Sanofi. D.K. is part of the DUB Alliance that includes Cancer Research Technology and FORMA Therapeutics, and is a consultant for FORMA Therapeutics. The other authors declare no competing financial interest.

