## [Peer Review File · Nature Communications]

Reviewers' comments:

Reviewer #3 (Remarks to the Author):

Two key and inter-related issues around the biochemical evidence supporting claims that CYLD directly deubiquitinates and stabilizes P53 have been more thoroughly addressed in the revised manuscript.

Firstly, the authors claim that CYLD interaction results in P53 stabilization. Figure 7c now shows enhanced accumulation of P53 in response to CpT on CYLD over-expression. Although experiments to formally prove that this is due to P53 stabilisation, rather than e.g. increased translation/transcription (cycloheximide chase, mRNA analysis), have not been attempted, given the critical regulation of P53 through protein stability this is probably a reasonable interpretation.

Secondly, it was unclear what type of ubiquitin chains were removed by CYLD to stabilise P53, given the specificity of CYLD for e.g. K63 (non-degradative) over K48 (degradative) chains. The authors have now undertaken a series of experiments to address this question. They show that over-expressed CYLD can remove from P53: (i) ubiquitin chains formed from K48-only or K63-only ubiquitin mutants, and (ii) chains formed by wild-type ubiquitin recognised by both K48 and K63-specific antibodies. They also cite an in vitro experiment in the rebuttal letter, these data should be included in the manuscript. They then go on to suggest CYLD is actually removing mixed/branched ubiquitin chains from P53, based on a UbiCREST experiment, such that the K63-activity of CYLD also removes some K48-linked ubiquitin. Whilst this is a plausible and interesting explanation, in their discussion the authors need to better reconcile their mixed/branched chain hypothesis with the ability of CYLD to remove K48-only ubiquitin chains in cells/in vitro.

Reviewer #5 (Remarks to the Author):

The authors have responded well to the concerns of the previous reviews and thus have substantially strengthened and improved the manuscript. It is now acceptable for publication.

Reviewer #6 (Remarks to the Author):

Fernandez-Majada et al

This paper provides good evidence that CYLD negatively regulates p53 levels/activity but the mechanisms involved are not clear (whether direct or indirect). The authors propose a pathway whereby CYLD, a known K63-targeting deubiquitinase, can remove both K48 and K63-linked ubiquitin chains from p53 leading to its stabilization and enhanced activity following DNA damage. Several problems with the manuscript are obvious:

--Loss of CYLD reduces p53 levels, but it is not addressed whether this could be at the level of RNA or truly at the level of protein destabilization (i.e., will a proteasome inhibitor stabilize p53 levels following CYLD loss/knockdown?).

--Loss of CYLD leads to upregulation of IKK and JNK signaling, both of which are known to downregulate p53. References to this effect were not included - for example, the work of Inder Verma has shown that IKK2 can phosphorylate and destabilize p53 (Y. Xia et al, PNAS 2009). There are several references regarding the effect of JNK on p53, and that loss of CYLD leads to upregulation of JNK. Thus, if knockout of CYLD function upregulates IKK and/or JNK then p53 levels should go down through established mechanisms. Inhibition of IKK2 and JNK should be tested to determine if p53 stabilization is regulated through upregulated IKK. The authors have

completely failed to address these potential mechanisms.

--The studies using *C. elegans* are misleading. The CEP-1 p53-like protein (the homology is largely limited to the transactivation and DNA-binding domains) is not regulated by ubiquitination but through control of translation. Consistent with this, *C. elegans* does not have MDM2. Thus when the authors claim that their findings prove that NF- κ B signaling is not involved in the ability of CYLD to regulate p53 (since worms don't have NF- κ B components), they are misleading readers since the proposed pathway of CYLD deubiquitinating p53 in mammalian cells doesn't exist in *C. elegans* -- thus they cannot make the claims they make. Additionally, they never show that levels of CEP-1 are regulated by CYLD in worms.

--In Fig.3g, in keratinocytes the induction of p53-target gene expression is the same with WT or mut CYLD in response to DNA damage, it is just that the baseline/starting point is different. This is not consistent with their model.

--In Figure 7 (7a, and others in this figure) CYLD expression didn't affect p53 levels. Why not?

--The finding that p53 can be ubiquitinated through K63 linkage is interesting but not fully explored. Can the authors show that a K63-specific ubiquitin antibody (H. Wang et al., PNAS 2008) recognizes p53 - endogenous p53 - stabilized with a proteasome inhibitor??

This reviewer believes that it is quite possible that the effect of CYLD on p53 is at two levels - whereby the loss of CYLD activates IKK and JNK to destabilize p53 and that CYLD removes K63-linked ubiquitination which has an uncharacterized function on p53.

Comments regarding previous review:

-I have no concerns with the tumor studies (other than that NF- κ B, JNK, WNT signaling needs to be analyzed). The authors find, as expected, that CYLD functions as a tumor suppressor. Nuances in tumor growth characteristics are just that and are a distraction from what is the critical point of the paper -- how CYLD is affecting p53 levels/activity - which is unclear.

-Based on comments provided above, I agree with the reviewers that the authors must examine IKK/NF- κ B signaling, as well as JNK and WNT, occurring with loss of CYLD as these pathways may directly modulate p53 (which is published).

-I agree with the 3rd reviewer that "there remains no biochemical evidence that CYLD interaction and deubiquitination result in p53 stabilization". See my comments above.

[my overall recommendation is rejection.]

Reviewers' comments:

Reviewer #3 (Remarks to the Author):

Two key and inter-related issues around the biochemical evidence supporting claims that CYLD directly deubiquitinates and stabilizes P53 have been more thorough addressed in the revised manuscript.

Firstly, the authors claim that CYLD interaction results in P53 stabilization. Figure 7c now shows enhanced accumulation of P53 in response to CpT on CYLD over-expression. Although experiments to formally prove that this is due to P53 stabilisation, rather than e.g. increased translation/transcription (cycloheximide chase, mRNA analysis), have not been attempted, given the critical regulation of P53 through protein stability this is probably a reasonable interpretation.

We had tested early on during the development of this project the possibility that CYLD might regulate p53 mRNA levels in intestinal organoids and primary keratinocytes in response to DNA damage induced by CpT treatment. As shown in figure 1 for reviewers below CYLD Δ 932 mutant cells did not show reduced p53 gene expression in response to CpT when compared to CYLD wild type cells. In contrast, p53 mRNA levels in CYLD Δ 932 mutant cells were mildly increased at 3-4h after treatment (Fig.3e,g). Therefore, the reduced p53 protein levels could not be explained by impaired p53 mRNA expression. We have not tested whether the rate of p53 mRNA translation is altered by the lack of CYLD catalytic activity. However, as the reviewer points out, based on our findings and the critical role of ubiquitination in regulating p53 protein stability, we believe our interpretation that CYLD regulates p53 protein stability by removing K48 ubiquitin chains (indirectly by cleaving K63 chains) is reasonable and supported by the presented data.

Figure 1 for reviewers. qRT-PCR analysis of TP53 gene expression in intestinal organoids (upper panel) or primary epidermal keratinocytes (lower panel) from CYLD Δ 932^{FL} and CYLD Δ 932^{epi} mice treated with CpT for the indicated time points.

Secondly, it was unclear what type of ubiquitin chains were removed by CYLD to stabilise P53, given the specificity of CYLD for e.g. K63 (non-degradative) over K48 (degradative) chains. The authors have now undertaken a series of experiments to address this question. They show that over-expressed CYLD can remove from P53: (i) ubiquitin chains formed from K48-only or K63-only ubiquitin mutants, and (ii) chains formed by wild-type ubiquitin recognised by both K48 and K63-specific antibodies. They also cite an *in vitro* experiment in the rebuttal letter, these data should be included in the manuscript.

Following the suggestion of the reviewer, we have now included in new Supplemental Figure 7e the results from the experiment showing that recombinant CYLD removes ubiquitin chains from p53 *in vitro*.

They then go on to suggest CYLD is actually removing mixed/branched ubiquitin chains from P53, based on a UbiCREST experiment, such that the K63-activity of CYLD also removes some K48-linked ubiquitin. Whilst this is a plausible and interesting explanation, in their discussion the authors need to better reconcile their mixed/branched chain hypothesis with the ability of CYLD to remove K48-only ubiquitin chains in cells/*in vitro*.

Following the suggestion of the reviewer we have modified the text (lines 408-415) to provide a more comprehensive discussion of our findings and our interpretation of the results.

Reviewer #5 (Remarks to the Author):

The authors have responded well to the concerns of the previous reviews and thus have substantially strengthened and improved the manuscript. It is now acceptable for publication.

Reviewer #6 (Remarks to the Author):

Fernandez-Majada et al

This paper provides good evidence that CYLD negatively regulates p53 levels/activity but the mechanisms involved are not clear (whether direct or indirect). The authors propose a pathway whereby CYLD, a known K63-targeting deubiquitinase, can remove both K48 and K63-linked ubiquitin chains from p53 leading to its stabilization and enhanced activity following DNA damage. Several problems with the manuscript are obvious:

--Loss of CYLD reduces p53 levels, but it is not addressed whether this could be at the level of RNA or truly at the level of protein destabilization (i.e., will a proteasome inhibitor stabilize p53 levels following CYLD loss/knockdown?).

Please see our response to the first comment of reviewer 3 above.

-Loss of CYLD leads to upregulation of IKK and JNK signaling, both of which are known to downregulate p53. References to this effect were not included - for example, the work of Inder Verma has shown that IKK2 can phosphorylate and destabilize p53 (Y. Xia et al, PNAS 2009). There are several references regarding the effect of JNK on p53, and that loss of CYLD leads to upregulation of JNK. Thus, if knockout of CYLD function upregulates IKK and/or JNK then p53 levels should go down through established mechanisms. Inhibition of IKK2 and JNK should be tested to determine if p53 stabilization is regulated through upregulated IKK. The authors have completely failed to address these potential mechanisms.

The reviewer wonders whether the impaired stabilization of p53 in cells lacking CYLD catalytic activity could be indirect via the upregulation of IKK and JNK signaling. While the JNK pathway has been reported to influence p53 stabilization and activation, these interactions seem to be complex with different reports providing conflicting results whether IKK and JNK signaling positively or negatively regulate p53 stability and activation. For example, several papers showed that activation of JNK signaling results in increased stabilization and activation of p53 (e.g. see (Fuchs et al., 1998; Kim and Shim, 2016; Reyes-Zurita et al., 2011; Topisirovic et al., 2009)). According to these reports, increased activation of JNK in CYLD-deficient cells should result in increased stability and activation of p53. Considering that we observed reduced stabilization and activation of p53 in CYLD-deficient cells, this could not be explained by an inhibitory effect of CYLD in JNK signaling. The role of IKK/NF- κ B signaling in the regulation of p53 is also not clear, as the two pathways seem to have multiple points of cross-regulation. However, our experiments showed that primary keratinocytes as well as intestinal organoids expressing the catalytically inactive mutant CYLD Δ 932 did not exhibit substantially increased NF- κ B or JNK activation in response to CpT (Supplementary Fig. 6), supporting that the decreased stabilization of p53 in CYLD-deficient cells is not due to elevated NF- κ B or JNK signaling.

We would also like to stress that we provide extensive experimental evidence that CYLD directly interacts and deubiquitinates p53. Specifically, by *in vivo* immunoprecipitation experiments using p53 and CYLD antibodies we have shown that CYLD and p53 interact in response to DNA damage in an overexpressed system and at the endogenous level (Figures 6 a, b, d, and e). Moreover, we showed that the interaction between CYLD and p53 is direct employing recombinant proteins in an *in vitro* GST pull-down assay (Figure 6c). Importantly, we also showed that recombinant His-CYLD reduced ubiquitination of p53 immunoprecipitated from DNA damage-treated HEK-293T cells, showing that CYLD directly removes ubiquitin chains from p53 in a cell-free *in vitro* assay (see new Supplementary Fig 7e in the revised manuscript). Collectively, these results show that in response to DNA damage CYLD directly binds and deubiquitinates p53 to induce its stabilization and activation.

--The studies using *C. elegans* are misleading. The CEP-1 p53-like protein (the homology is largely limited to the transactivation and DNA-binding domains) is

not regulated by ubiquitination but through control of translation. Consistent with this, *C. elegans* does not have MDM2. Thus when the authors claim that their findings prove that NF- κ B signaling is not involved in the ability of CYLD to regulate p53 (since worms don't have NF- κ B components), they are misleading readers since the proposed pathway of CYLD deubiquitinating p53 in mammalian cells doesn't exist in *C. elegans* -- thus they cannot make the claims they make. Additionally, they never show that levels of CEP-1 are regulated by CYLD in worms.

With all respect to the reviewer's opinion, we do not agree that our *C. elegans* studies presented in the manuscript are misleading. Indeed, the sequence homology between CEP-1 and human p53 is confined to the transactivation and DNA-binding domain as we and others have previously determined (Derry et al., 2001; Schumacher et al., 2001). Importantly, we have previously demonstrated that CEP-1 can recognize p53 consensus sites similarly to human p53 (Schumacher et al., 2001) and controls the expression of BH3 only domain proteins in response to DNA damage (Schumacher et al., 2005b) indicating that CEP-1 and p53 operate in a highly functionally conserved manner. While it is correct that we have previously established that the induction of CEP-1 is regulated by the translational control of *cep-1* mRNA through GLD-1 (Schumacher et al., 2005a), this does not mean that the stability of CEP-1 couldn't be additionally controlled by ubiquitin-dependent protein degradation. Indeed, even though BLAST searches have not revealed an MDM2 orthologue, the SCF^{F^{SN}-1} ubiquitin ligase was shown to regulate CEP-1-mediated responses suggesting that ubiquitination does control CEP-1/p53 activation also in *C. elegans* (Gao et al., 2008). Therefore, ample evidence exist that CEP-1 is a functional p53 homolog and that ubiquitin-dependent protein degradation may regulate also CEP-1 protein levels in *C. elegans*. However, since the precise mechanism by which ubiquitination regulates CEP-1/p53 in *C. elegans* remains poorly understood, we have modified the discussion of our *C. elegans* results to reflect this point and provide a more balanced interpretation of our findings.

--In Fig.3g, in keratinocytes the induction of p53-target gene expression is the same with WT or mut CYLD in response to DNA damage, it is just that the baseline/starting point is different. This is not consistent with their model.

We are not sure if the reviewer refers to Figure 3g (immunoblots) or Figure 3h (qRT-PCR analysis of mRNA levels). Regarding Figure 3g, we have analyzed the expression of p53 and p21 in response to genotoxic stress in primary keratinocytes prepared from different control and CYLD Δ 932 mutant mice in four independent experiments. We found consistently that p53 stabilization and p21 expression were reduced in CYLD Δ 932 keratinocytes compared to control cells (see Figure 2a for reviewers below which includes the blot shown in figure 3g as well as three additional blots). These results are consistent with our findings in intestinal organoids, where lack of CYLD catalytic activity also resulted in decreased stabilization of p53 and impaired induction of p21 (Fig. 3e and Fig. 4a).

The results presented in Figure 3h show that the expression of two p53 target genes (Cdkn1a and Bax) is reduced in CYLD mutant keratinocytes after DNA damage but

also at basal levels. In our view, the important result of this experiment is that p53-dependent gene expression is impaired in the absence of CYLD catalytic activity. In this experiment we do not consider that the calculation of fold-induction compared to untreated cells gives an accurate assessment of the findings, on the contrary we believe that it is more important to refer to the overall expression levels. The 'basal' expression of these genes is likely to also be induced by low level p53 activation in these primary keratinocyte cultures (perhaps induced by the culture conditions), therefore the reduced basal levels observed in CYLD mutant cells would also be consistent with our model that CYLD catalytic activity is required for optimal p53 activation.

Figure 2 for reviewers. (a-d) Replicate experiments of immunoblot analysis of p53, p21 and Tubulin/Actin in primary epidermal keratinocytes prepared from the indicated mouse lines after treatment with CpT (a-c) or MMC (d) for the depicted time points.

--In Figure 7 (7a, and others in this figure) CYLD expression didn't affect p53 levels. Why not?

The experiments shown in figure 7a, b, d, e, f and g, are performed employing HEK-293T or HCT116 cells overexpressing p53 and treated with the proteasome inhibitor MG-132 in order to assess the effect of CYLD expression on the ubiquitination status of

p53. Therefore, this experimental setup cannot evaluate the effect of CYLD overexpression on p53 protein levels. We would like to draw the reviewer's attention to figure 7c were we treated HCT116 (expressing or not HA-CYLD) with a time course of CpT in the absence of p53 overexpression and without proteasome inhibition, and checked for p53 stabilization by immunoblot. Our results showed that CYLD overexpression indeed resulted in enhanced p53 stabilization in response to DNA damage. These results are in line with our conclusion that CYLD deubiquitinates p53 thus inducing its optimal stabilization and activation.

--The finding that p53 can be ubiquitinated through K63 linkage is interesting but not fully explored. Can the authors show that a K63-specific ubiquitin antibody (H. Wang et al., PNAS 2008) recognizes p53 - endogenous p53 - stabilized with a proteasome inhibitor??

We agree with the reviewer that the role of K63-linked ubiquitination of p53 needs to be explored further and we are planning future experiment to address the functional role of these chains in the regulation of p53 stabilization and activation. We would also like to point out that a previous study showed that Ubc13 mediated K63-linked p53 ubiquitination (Laine et al., 2006), therefore there is already evidence in the literature that p53 is decorated with K63-linked Ub chains. Furthermore, using linkage specific antibodies we show in figure 7f of the manuscript that overexpressed p53 is ubiquitinated with both K63 and K48-linked poly-ubiquitin chains and that CYLD is able to reduce both types of ubiquitin chains from p53. These results clearly show that p53 is indeed decorated with K63 chains. While examining the ubiquitin chain linkages in endogenous p53 would also be important, we respectfully suggest that these technically challenging experiments are outside of the scope of the current manuscript and can be performed in the context of our future studies of the role of K63 chains on p53 regulation.

This reviewer believes that it is quite possible that the effect of CYLD on p53 is at two levels - whereby the loss of CYLD activates IKK and JNK to destabilize p53 and that CYLD removes K63-linked ubiquitination which has an uncharacterized function on p53.

Please see above our response to the comment of the reviewer on the potential role of IKK and JNK in the observed regulation of p53 stabilization by CYLD.

Comments regarding previous review:

-I have no concerns with the tumor studies (other than that NF-kB, JNK, WNT signaling needs to be analyzed). The authors find, as expected, that CYLD functions as a tumor suppressor. Nuances in tumor growth characteristics are just that and are a distraction from what is the critical point of the paper -- how CYLD is affecting p53 levels/activity - which is unclear.

-Based on comments provided above, I agree with the reviewers that the authors must examine IKK/NF- κ B signaling, as well as JNK and WNT, occurring with loss of CYLD as these pathways may directly modulate p53 (which is published).

-I agree with the 3rd reviewer that "there remains no biochemical evidence that CYLD interaction and deubiquitination result in p53 stabilization". See my comments above.

[my overall recommendation is rejection.]

References

Derry, W.B., Putzke, A.P., and Rothman, J.H. (2001). *Caenorhabditis elegans* p53: role in apoptosis, meiosis, and stress resistance. *Science* 294, 591-595.

Fuchs, S.Y., Adler, V., Pincus, M.R., and Ronai, Z. (1998). MEKK1/JNK signaling stabilizes and activates p53. *Proc Natl Acad Sci U S A* 95, 10541-10546.

Gao, M.X., Liao, E.H., Yu, B., Wang, Y., Zhen, M., and Derry, W.B. (2008). The SCF FSN-1 ubiquitin ligase controls germline apoptosis through CEP-1/p53 in *C. elegans*. *Cell Death Differ* 15, 1054-1062.

Kim, J., and Shim, M. (2016). COX-2 inhibitor NS-398 suppresses doxorubicin-induced p53 accumulation through inhibition of ROS-mediated Jnk activation. *Mol Carcinog*.

Laine, A., Topisirovic, I., Zhai, D., Reed, J.C., Borden, K.L., and Ronai, Z. (2006). Regulation of p53 localization and activity by Ubc13. *Mol Cell Biol* 26, 8901-8913.

Reyes-Zurita, F.J., Pachon-Pena, G., Lizarraga, D., Rufino-Palomares, E.E., Cascante, M., and Lupianez, J.A. (2011). The natural triterpene maslinic acid induces apoptosis in HT29 colon cancer cells by a JNK-p53-dependent mechanism. *BMC Cancer* 11, 154.

Schumacher, B., Hanazawa, M., Lee, M.H., Nayak, S., Volkmann, K., Hofmann, E.R., Hengartner, M., Schedl, T., and Gartner, A. (2005a). Translational repression of *C. elegans* p53 by GLD-1 regulates DNA damage-induced apoptosis. *Cell* 120, 357-368.

Schumacher, B., Hofmann, K., Boulton, S., and Gartner, A. (2001). The *C. elegans* homolog of the p53 tumor suppressor is required for DNA damage-induced apoptosis. *Curr Biol* 11, 1722-1727.

Schumacher, B., Schertel, C., Wittenburg, N., Tuck, S., Mitani, S., Gartner, A., Conradt, B., and Shaham, S. (2005b). *C. elegans* ced-13 can promote apoptosis and is induced in response to DNA damage. *Cell Death Differ* 12, 153-161.

REVIEWERS' COMMENTS:

Reviewer #3 (Remarks to the Author):

The authors include an additional experiment in their rebuttal letter suggesting the effect of CYLD on P53 levels is unlikely to be mediated at the mRNA level, and have clarified their discussion of the mixed/branched chain hypothesis.

Reviewer #6 (Remarks to the Author):

The authors have addressed most of my concerns. They ignored my request to show that proteasome inhibitor treatment would rescue the destabilization of p53 upon Cyld inactivation. This is a key experiment that is missing, and that would link Cyld with K48 deubiquitination (with the expected proteasome-dependent degradation).

They failed to reference a key paper (Laine et al, MCB 2006) which shows that p53 is K63-ubiquitination. Interestingly, the authors referenced this in their response to reviewers but did not include the reference in the text.

Reviewer's comments

Reviewer #3 (Remarks to the Author):

The authors include an additional experiment in their rebuttal letter suggesting the effect of CYLD on P53 levels is unlikely to be mediated at the mRNA level, and have clarified their discussion of the mixed/branched chain hypothesis.

We would like to thank the reviewer for their positive comments on our revised manuscript.

Reviewer #6 (Remarks to the Author):

The authors have addressed most of my concerns. They ignored my request to show that proteasome inhibitor treatment would rescue the destabilization of p53 upon Cyld inactivation. This is a key experiment that is missing, and that would link Cyld with K48 deubiquitination (with the expected proteasome-dependent degradation).

We are glad that our response addressed most of the reviewer's concerns. Regarding the suggested experiment testing whether proteasome inhibition would stabilize p53 in CYLD mutant cells, we believe that our response to reviewers 3 and 6 in the previous revision of the paper addressed this issue. As we found that CYLD mutation did not decrease the mRNA levels of p53, we believe that our interpretation that CYLD regulates p53 protein stability is reasonable and supported by our results. Besides, as shown in numerous previous studies proteasome inhibition universally increases p53 levels therefore, particularly considering that CYLD is not the only regulator of p53 protein stability, it would be difficult to assign any result to the function of CYLD. We therefore respectfully suggest that this experiment is not essential to support our conclusions.

They failed to reference a key paper (Laine et al, MCB 2006) which shows that p53 is K63-ubiquitination. Interestingly, the authors referenced this in their response to reviewers but did not include the reference in the text.

We regret we did not reference this paper in our manuscript and we thank the reviewer for pointing this out. We have now cited this reference in the main text of the revised manuscript.